# Diverse redox-mediated transformations to realize the *para*-quinoid, σ-bond, and *ortho*-diphenoquinoid forms

Takashi Harimoto [1,2] ✉, Moto Kikuchi[1], Takanori Suzuki [1] & Yusuke Ishigaki [1] ✉

π-Electron systems with multiple redox-active units have attracted attention in various fields due to their potential applications. However, the design strategy remains elusive to selectively synthesize the diverse molecular structures of redox-convertible species. In this study, covalently linked quinodimethane derivatives with a sulfur bridge [$(Ar_4QD)_2S$] were designed as redox-active motifs that can be converted into three different geometries via redox reaction. Here we show that the favored geometry of the corresponding redox states of $(Ar_4QD)_2S$ can be precisely controlled by adjusting the steric bulk of the substituents on the aryl group to change the proximity of the quinodimethane units. Notably, this redox-mediated strategy also leads to the isolation and structural determination of the missing link with an *o*-diphenoquinoid structure, a diphenoquinoid isomer whose isolation had remained elusive for almost a century. Thus, this study provides a method that allows the modulation/control of electronically and/or thermodynamically stable structures, as well as their electronic and spectroscopic properties.

Recently, π-electron systems composed of multiple redox-active units (electrophores) have attracted attention in various fields, including synthetic chemistry, electronics, and life sciences[1–4]. Notably, the precise assembly of electrophores gives them advanced properties, such as the capacity to transport multiple electrons[5–8], electrochemical amphotericity[9–11], and supramolecular functionality[12–15]. Despite those attractive properties, the corresponding charged states of these multi-electrophore systems generally inherit the structural characteristics of each original electrophore without the formation of π- or σ-bonds between the neighboring electrophores. Therefore, it is still a challenge to selectively synthesize the diverse molecular structures of redox-convertible species in multi-electrophore systems.

*Para*-quinodimethane (*p*-QD), a typical cross-conjugated skeleton, has been used for several decades as a motif capable of inducing reversible electron transfer[16–25]. This process is facilitated by a biradical contribution with aromatization of the six-membered ring, which allows the generation of stable charged states via electron transfer

(Fig. 1a). Therefore, in QD-based redox systems, methods for controlling the absorption properties, operating potential, and even the reversibility of interconversion have been well studied[18,26–36]. However, for the multiple *p*-QD systems, molecular structures that appear in a specific redox state are strictly limited to those based on the simple accumulation of charge in the original QD frameworks (Fig. 1b). Thus, the establishment of a molecular design strategy that can diversify the structures of charged states with precise control of the most favored geometry of a specific redox state is needed.

To meet this need, we envisaged that the direct connection of two QD electrophores via a C-C covalent bond could allow us to expand the structural diversity of the redox states. Specifically, the direct connection of two QD units could induce the recombination of chemical bonds between the two radical cation units, which are transiently generated by the one-electron (1e) oxidation of each QD electrophore, resulting in the isolation of various structures, e.g., *p*-quinoid form **A**, σ-bond form **B**, and *o*-diphenoquinoid form **C** (Fig. 1c), the latter two of

[1]Department of Chemistry, Faculty of Science, Hokkaido University, Sapporo, Japan. [2]Present address: Institute for Molecular Science, Myodaiji, Okazaki, Japan. ✉e-mail: t-harimoto@ims.ac.jp; yishigaki@sci.hokudai.ac.jp

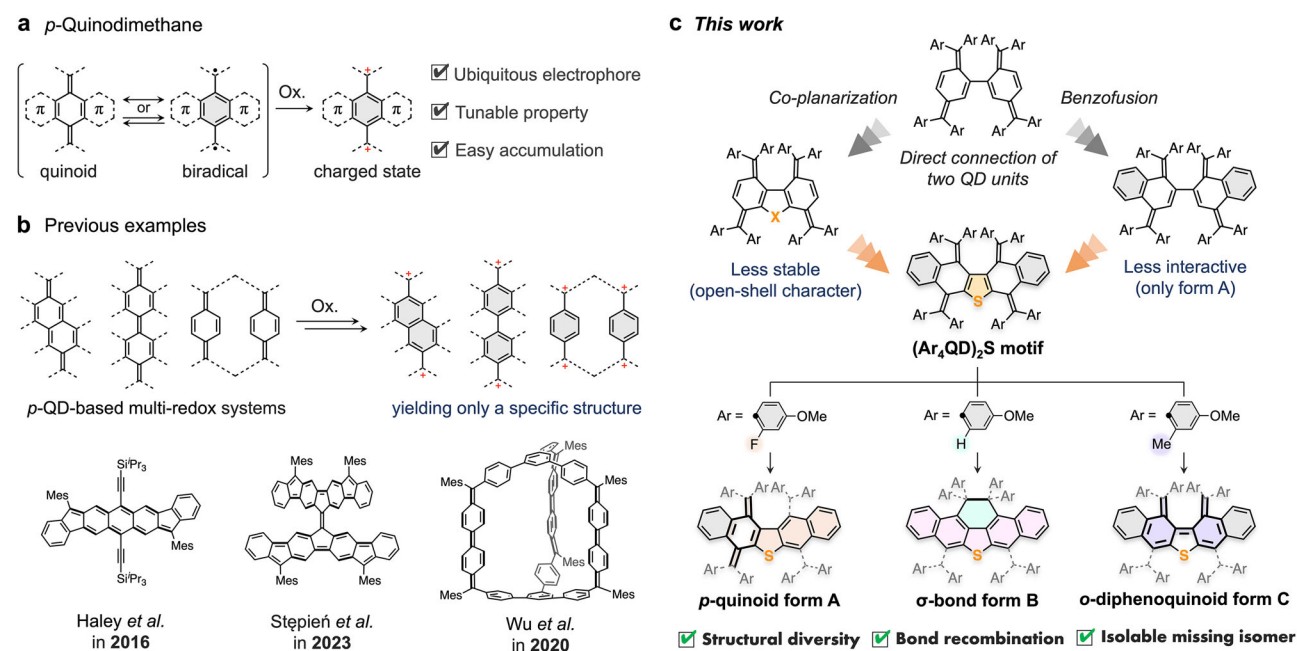

**Fig. 1 | Design concept for this study. a** *p*-Quinodimethane (*p*-QD) motifs. **b** Previous examples: *p*-QD systems with multi-redox behavior. **c** This work: structural diversification of *p*-QDs using (Ar₄QD)₂S motifs.

which are not the simple accumulated structures of *p*-QD-based redox systems. Among them, form **C** is a missing isomer of the diphenoquinoid structure, even though compounds with a diphenoquinoid structure had been introduced nearly a century ago, and can therefore be considered to be a "missing" link. Chichibabin reported diphenoquinoid hydrocarbons over a century ago[37], and ever since, a variety of precisely designed π-electron systems with a *p*-diphenoquinoid structure have been isolated and studied experimentally and theoretically[28,31,38–42]. Following a series of investigations on ditolyl derivatives by Chichibabin et al.[43], Wittig et al. tried to pursue the isolation of compounds with an *o*-diphenoquinoid structure in 1933[44], albeit that their isolation was impossible due to the preferential formation of biradical- and then σ-bond valence tautomers. Thus, the synthetic method developed here should provide a chance to delve deeper into the characteristics of *o*-diphenoquinoid species. Density functional theory (DFT) calculations suggested that both the co-planarization of the two QD units and the fusion of two benzene rings (benzofusion) to a central π-system could also be an effective way to guarantee the formation of *o*-diphenoquinoid form **C** (*cf.* p.S52). Based on these considerations, we designed and synthesized sulfur-bridged closed-shell π-electron systems **1** with two extra benzene rings [(Ar₄QD)₂S] (Fig. 1c). As the rational stabilization/destabilization of a structure is key to creating structural diversity, we aimed to control the structural preferences via the substituent effects on the aryl groups in combination with the proximity of the QD units.

## Results and discussion
### Synthesis of (Ar₄QD)₂S 1
To construct the covalently connected QD-based redox systems with a sulfur bridge, we adopted the ring-contraction strategy shown in Fig. 2a. We selected a 4-methoxy-substituted phenyl group as the aryl group, given that it has enough electron-donating character to stabilize both the cationic states and the neutral state. Following an eightfold Suzuki-Miyaura cross-coupling reaction, octaarylated sulfoxide **4a** was obtained in 48% yield from tetrakis(dibromomethylene) precursor **3**, which was oxidatively prepared from dithiin precursor **2**[45]. The targeted (Ar₄QD)₂S **1a** (Ar = 2-fluoro-4-methoxyphenyl) was prepared in

94% yield from sulfoxide **4a** via a desulfoxidation reaction upon heating to 413 K in DMSO.

In addition, to control the molecular structure induced by redox reactions, the steric bulk of the substituents on the aryl groups was varied. Thus, we prepared (Ar₄QD)₂S analogs **1b** (58%) and **1c** (83%) with a hydrogen atom and a methyl group at the 2-position of the 4-methoxyphenyl groups, respectively, under the comparable thermal conditions. Notably, the formation of the overcrowded exomethylene moieties via a desulfoxidation reaction after the introduction of the aryl groups is feasible even in the case of most congested methyl-substituted **1c**.

Single-crystal X-ray diffraction analyses revealed that all of these (Ar₄QD)₂S species contain a central pentacyclic skeleton that adopts a zigzag conformation with the two folded QD units arranged so that their concave surfaces are facing in opposite directions (Fig. 2b–d, Tables S1–S3). The two exomethylene moieties opposite the sulfur atom of the thiophene ring create a highly congested geometry, and the distance between the two exomethylene carbons C1 and C2 is 3.773(3) Å for **1a**, 3.656(2) Å for **1b**, and 3.845(9) Å for **1c**. The C1–C2 distance increases with increasing bulk of the substituents on the aryl groups. This distance is significantly shorter than the previously reported bis-quinodimethane derivatives, i.e., 5,7,12,14-tetrakis(diarylmethylene)tetrahydropentacenes, in which the two QD units are bridged by a six-membered ring (C···C distance of ~5.9 Å)[46]. These results indicate that the proximity of the two QD units affects the redox behavior and the structure of the oxidized species.

### Isolation of dicationic species 1a²⁺ with *p*-quinoid form A
To investigate the redox properties in detail, we conducted cyclic-voltammetry experiments in CH₂Cl₂ (Fig. 3a). For **1a**, the voltammogram showed that two-stage two-electron (2e) processes occur in both the oxidation and reduction waves ($E_{peak}^{ox1} = +1.00$ V vs. SCE and $E_{peak}^{ox2} = +1.21$ V for **1a**; $E_{peak}^{red1} = +1.06$ V and $E_{peak}^{red2} = +0.87$ V for **1a⁴⁺**). The presence of a 2e process for each oxidation wave was verified using ferrocene as an external standard and confirmed using differential pulse voltammetry (DPV) (Fig. S15). Such a quasi-reversible redox peak stands in contrast to the dynamic redox behavior observed for an anthraquinodimethane derivative with the

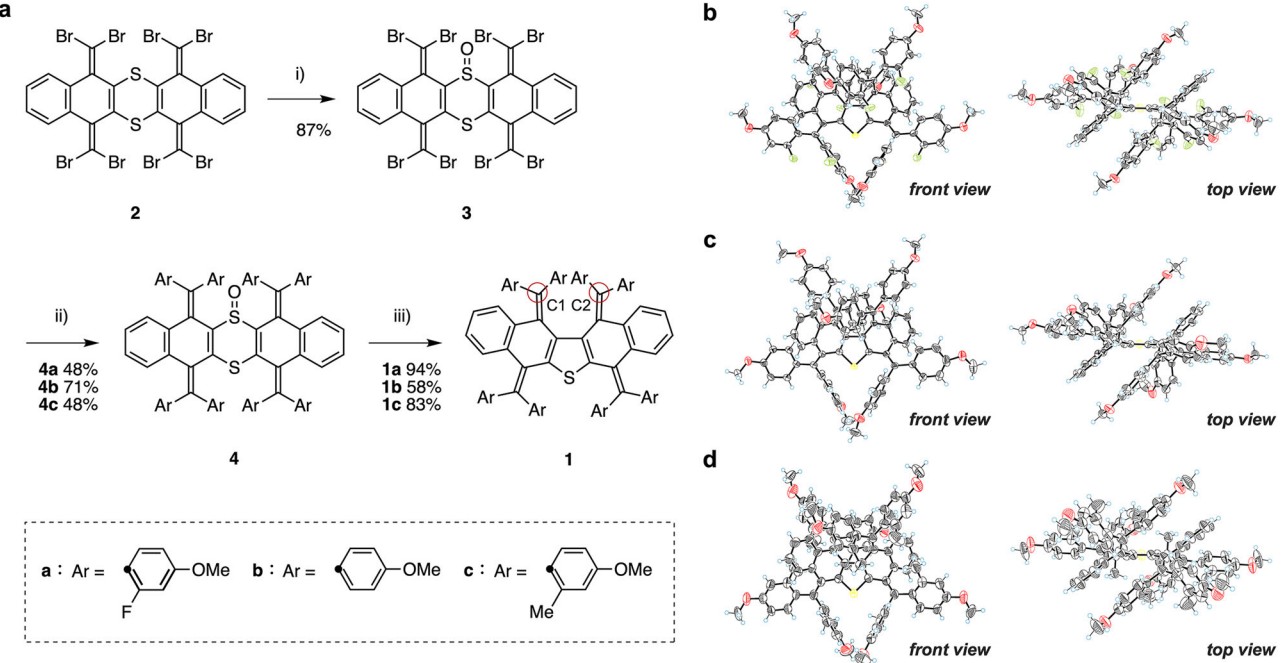

**Fig. 2 | Synthetic and X-ray crystallographic details of the neutral (Ar₄QD)₂S.** **a** Preparation of newly designed (Ar₄QD)₂S **1**. (i) H₂O₂ in CHCl₃:AcOH (3:1), (ii) ArB(OH)₂, K₂CO₃, and Pd(PPh₃)₄ in toluene:EtOH:H₂O (10:1:1), (iii) heating in DMSO for **1a** and **1c**, or neat under reduced pressure for **1b**. X-ray crystal structures (ORTEP drawings) of (**b**) **1a**, (**c**) **1b**, and (**d**) **1c** with thermal ellipsoids at 50% probability. Color code: atoms, C: gray, O: red, F: light green, S: yellow, H: light blue.

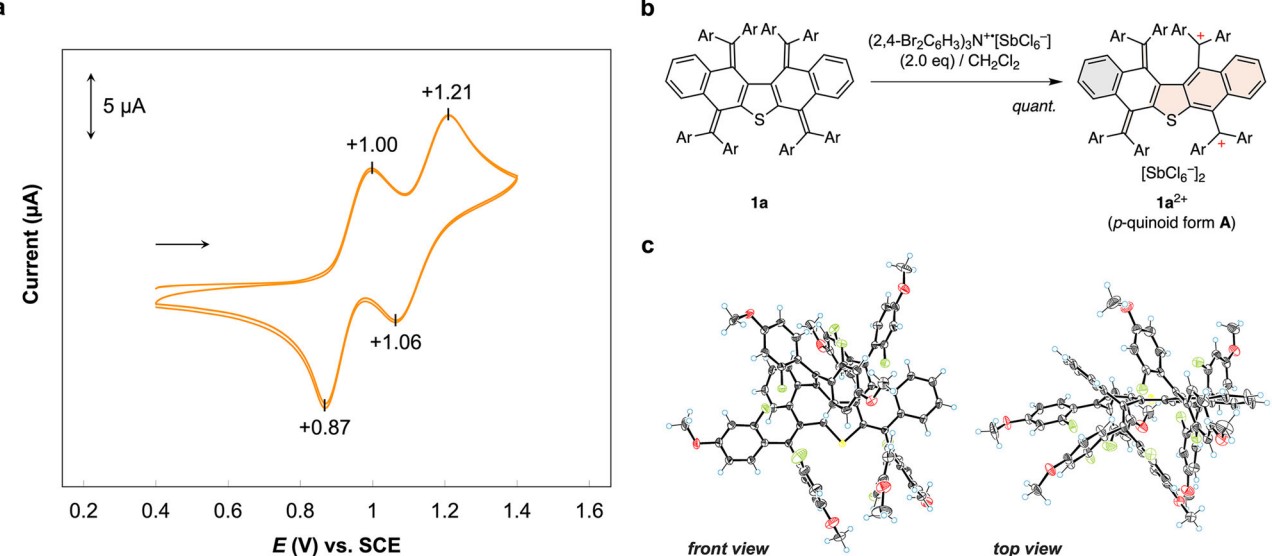

**Fig. 3 | Redox behavior of fluorine-substituted derivative 1a.** **a** Cyclic voltammogram of **1a** at 298 K in CH₂Cl₂ containing 0.1 M [Bu₄N⁺][BF₄⁻] as the supporting electrolyte (scan rate: 0.1 V s⁻¹; Pt electrode). **b** Preparation of dication salt **1a²⁺**[SbCl₆⁻]₂. **c** X-ray crystal structures (ORTEP drawings) of **1a²⁺**[SbCl₆⁻]₂ determined at 100 K. Counterions, solvent molecules, and disordered atoms are omitted for clarity. Thermal ellipsoids are shown at 50% probability. Color code: atoms, C: gray, O: red, F: light green, S: yellow, H: light blue.

same 2-fluoro-4-methoxylphenyl groups. A large separation of the redox peaks is observed for the anthraquinodimethane derivative ($E_{peak}^{ox}$ = +1.30 V and $E_{peak}^{red}$ = +0.71 V; Fig. S16) because it has a different structural preference for a folded geometry in the neutral state and a twisted geometry in the dication state. As the electrochemical measurements suggested that the neutral or dicationic species of **1a** would adopt a geometry similar to its dicationic or neutral state, we aimed to isolate the dication salt of (Ar₄QD)₂S **1a** and determine its structure.

Upon treatment of **1a** with two equivalents of (2,4-Br₂C₆H₃)₃N⁺•[SbCl₆⁻], dication salt **1a²⁺**[SbCl₆⁻]₂ was obtained quantitatively (Fig. 3b). A single-crystal X-ray diffraction analysis revealed that dication **1a²⁺** adopts *p*-quinoid form **A**, in which a tetrakis(2-fluoro-4-methoxyphenyl)quinodimethane unit is annulated to the naphthothiophene skeleton with two diarylmethylium units (Fig. 3c, Table S6). *p*-Quinoid form **A** is common among previously reported dications based on an all-hexagon bisquinodimethane structure, in which dication **1a²⁺** can acquire stabilization energy by the formation of two

**a**

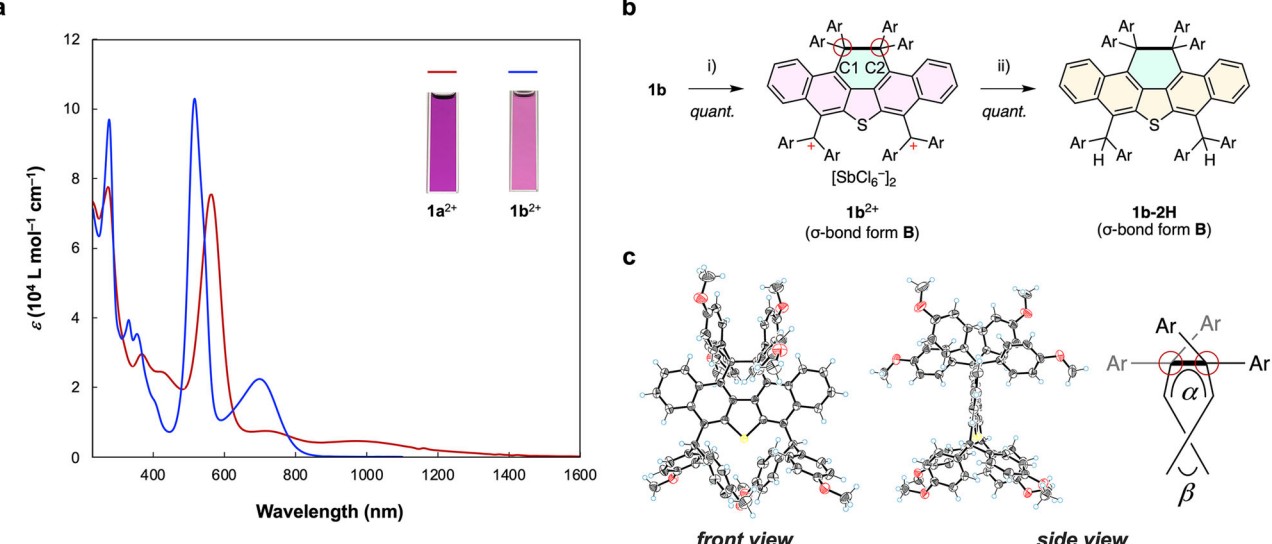

**Fig. 4 | Redox behavior of *ortho*-nonsubstituted derivative 1b. a** UV−vis−NIR spectra of **1a²⁺**[SbCl₆⁻]₂ (red), and **1b²⁺**[SbCl₆⁻]₂ (blue) in CH₂Cl₂. **b** Preparation of hydride adduct **1b-2H**. (i) Magic blue (2.0 eq) in CH₂Cl₂, (ii) NaBH₄ in MeCN. **c** X-ray crystal structures (ORTEP drawings) of **1b-2H** determined at 200 K. Solvent molecules are omitted for clarity. Thermal ellipsoids are shown at 50% probability. The dihedral angles $\alpha$ and $\beta$ are defined by four carbon atoms around the elongated C−C bond and within the thiophene ring, respectively. Color code: atoms, C: gray, O: red, S: yellow, H: light blue.

exomethylene π-bonds in the *p*-quinoid unit[46]. The formation of two diarylmethylium units on a (hetero)acene skeleton is often accompanied by a drastic structural change, which induces a major potential shift. However, the highly hindered structure presented here suppresses such a structural change when **1a** is converted to **1a²⁺**, thus resulting in only a minor potential shift.

**Investigation of dicationic species 1b²⁺ with σ-bond form B**

In search of a different geometry to that of *p*-quinoid form **A**, we turned our attention to the structure of dication **1b²⁺**, a species with 4-methoxyphenyl groups, which exert lower degrees of steric hindrance than the 2-fluoro-4-methoxyphenyl groups of **1a²⁺**. The target dication salt **1b²⁺**[SbCl₆⁻]₂ was successfully generated and isolated quantitatively by treatment of the neutral donor **1b** with two equivalents of (4-BrC₆H₄)₃N⁺•[SbCl₆⁻] (magic blue). The ¹H NMR spectrum of **1b²⁺**[SbCl₆⁻]₂ at 296 K in CD₃CN showed two sharp signals corresponding to the methoxy protons (δ = 3.68 and 4.16 ppm; Fig. S9) and these resonances were assigned to a closed-shell species with $C_{2v}$-symmetry. This suggests that **1b²⁺** adopts a structure different from that of **1a²⁺**, for which a much higher number of NMR signals was observed due to lower symmetry of the structure of *p*-quinoid form **A** (Fig. S8). To obtain further information regarding the structure of dication **1b²⁺**, we used UV−vis−NIR spectroscopy. In contrast to the neutral state, which absorbs only in the UV region, dications **1a²⁺**[SbCl₆⁻]₂ and **1b²⁺**[SbCl₆⁻]₂ exhibit a strong absorption in the visible region [$\lambda_{max}$/nm (log $\varepsilon$) in CH₂Cl₂: 562 (4.88) for **1a²⁺** and 515 (5.01) for **1b²⁺**] (Fig. 4a). These strong absorptions are characteristic of bis(methoxyphenyl)methylium-based chromophores.

In addition to the bands in the visible region, dications **1a²⁺** and **1b²⁺** also exhibit NIR absorptions (Fig. 4a). For **1a²⁺** with *p*-quinoid form **A**, a broad NIR absorption band [$\lambda_{max}$/nm (log $\varepsilon$) 970 (3.65)] extending to 1400 nm was observed. Based on time-dependent (TD)-DFT calculations at the CAM-B3LYP-D3/6−31G* level (Figs. S42, S46), this band was assigned to an intramolecular charge-transfer transition from the *p*-quinoid unit to the electron-deficient diarylmethylium units. In comparison, in the case of **1b²⁺**, the NIR absorption band is significantly blue-shifted relative to that of **1a²⁺** and is accompanied by an increase in the corresponding molar extinction coefficient $\varepsilon$ [$\lambda_{max}$/nm (log $\varepsilon$) 698 (4.35)]. This would be caused by the difference in the HOMO level,

indicating that **1b²⁺** has less donating π-skeleton such as heteroacene moiety in the σ-bond form **B**. These results suggest that dication **1b²⁺** adopts a structure that is substantially different from the *p*-quinoid form **A** of **1a²⁺**.

The X-ray diffraction analysis of **1b²⁺**[SbCl₆⁻]₂ revealed that dication **1b²⁺** does not adopt *p*-quinoid form **A** but σ-bond form **B** (Fig. S43), in which an elongated C($sp^3$)-C($sp^3$) single bond between the two diarylmethylene carbons C1 and C2 was observed in both of the two crystallographically independent molecules (mol-1/mol-2) per unit cell [1.731(11) Å for mol-1 and 1.709(9) Å for mol-2 at 100 K; typical length of a C($sp^3$)-C($sp^3$) bond: 1.54 Å].

To evaluate the relative energies ($E_{rel}$) between *p*-quinoid form **A** and σ-bond form **B** of the dications **1²⁺**, DFT calculations were performed at the CAM-B3LYP-D3/6-31 G* level. The results showed that the **A** and **B** forms obtained for both dications **1a²⁺** and **1b²⁺** have energy-minimized structures with twisted diarylmethylium units (Figs. S27, S28, Table S13). For **1a²⁺**, *p*-quinoid form **A** was calculated to be the most stable structure, while σ-bond form **B** is a metastable structure ($E_{rel}$: +5.17 kcal/mol). Conversely for **1b²⁺**, the relative stability of the two forms is reversed, and form **B** was calculated to be by 1.94 kcal/mol more stable than form **A**. Thus, based on an approach that balances the proximity and steric repulsion between the electrophores, the structural preference of the dications can be tuned. The σ-bond length in the optimized structure of form **B** was calculated to be 1.747 Å for **1a²⁺**, which is much longer than the calculated value for **1b²⁺** (1.680 Å). This is due to the efficient steric repulsion between the fluorine atoms in **1a²⁺**.

To obtain more accurate geometrical data by X-ray diffraction analysis while excluding charge effects, we decided to isolate a neutral species, namely the hydride adduct of dication **1b²⁺**, as this would allow a more detailed discussion of the reasons behind the elongated C($sp^3$)-C($sp^3$) single bond. The target hydride adduct **1b-2H** was successfully generated by treatment of dication **1b²⁺** with an excess of NaBH₄ (96% yield; Fig. 4b). It is of great importance for synthetic perspective because this result shows that the charged species generated by redox stimulation can be converted to a neutral form, while maintaining the π-extended acene-like framework. According to the X-ray diffraction analysis of **1b-2H** at 200 K, the torsion angle $\alpha$ around the C($sp^3$)−C($sp^3$) bond is 27.74(18)° and the dihedral angle $\beta$ for the fused core is only

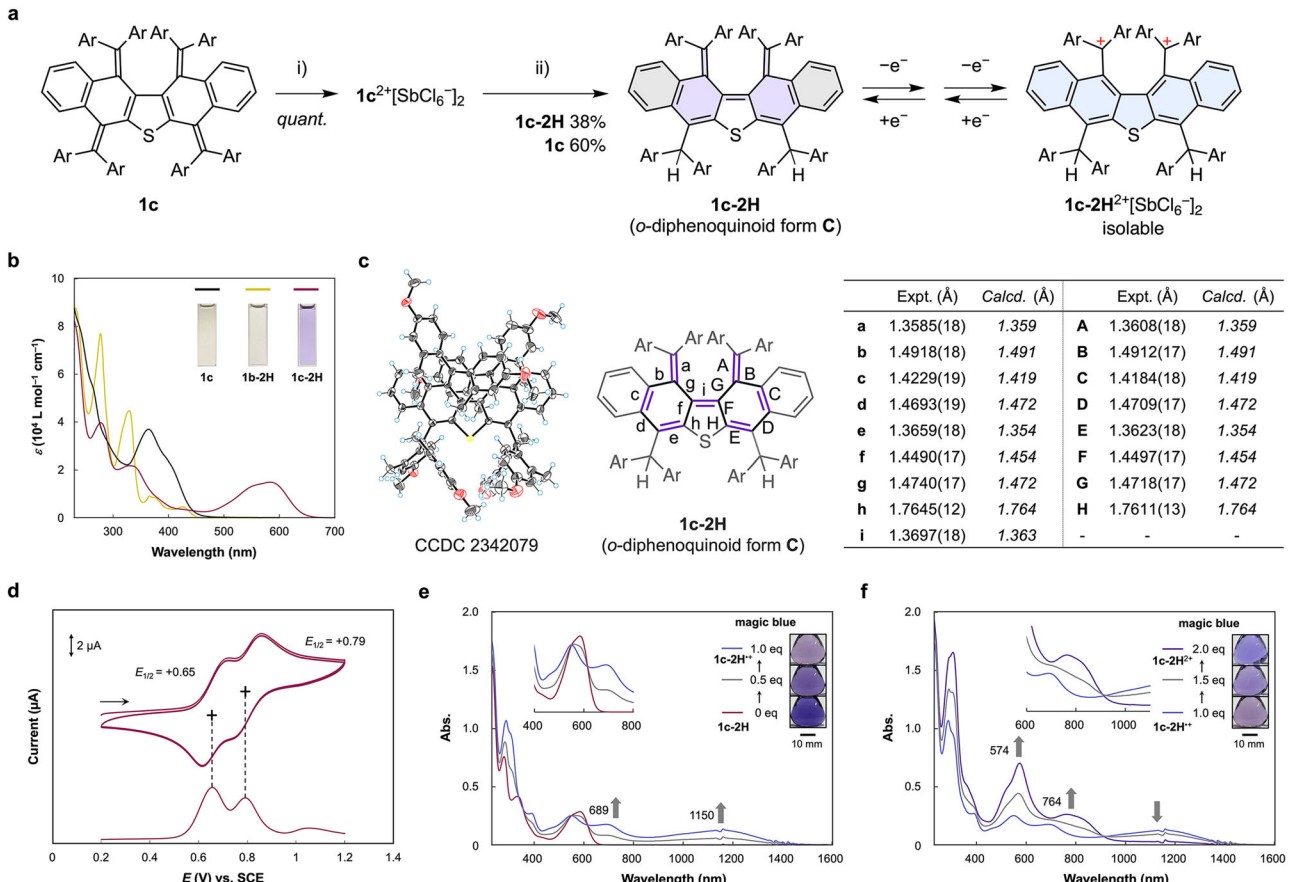

**Fig. 5 | Redox behavior of methyl-substituted derivative 1c and its derivative 1c-2H. a** Preparation of hydride adduct **1c-2H** via dication salt **1c$^{2+}$[SbCl$_6$$^-$]$_2$**. (i) Magic blue (2.0 eq) in CH$_2$Cl$_2$, (ii) NaBH$_4$ in MeCN. Redox interconversion between **1c-2H** and **1c-2H$^{2+}$[SbCl$_6$$^-$]$_2$** via intermediary radical cation. **b** UV–vis spectra of **1c** (black), **1b-2H** (yellow), and **1c-2H** (violet) in CH$_2$Cl$_2$. **c** X-ray crystal structures (ORTEP drawings) of **1c-2H** determined at 150 K. Solvent molecules are omitted for clarity. Thermal ellipsoids are shown at 50% probability. The bond lengths are shown in the accompanying table. The calculated values are shown in italics. Color code: atoms, C: gray, O: red, S: yellow, H: light blue. **d** Electrochemical measurements conducted on **1c-2H** at 298 K in CH$_2$Cl$_2$ containing 0.1 M [Bu$_4$N$^+$][BF$_4$$^-$] as the supporting electrolyte. Change in the UV–vis–NIR spectrum from (**e**) **1c-2H** (19.2 μM) to **1c-2H$^{+}$** (1st stage) and (**f**) from the as-prepared **1c-2H$^{+}$** to **1c-2H$^{2+}$** (2nd stage) upon adding several aliquots of magic blue in CH$_2$Cl$_2$.

6.45(5)° (Fig. 4c). Thanks to the high coplanarity of the dinaphthothiophene backbone and the clothespin effect (also called the scissor effect)[47] induced by the sulfur bridge, the C($sp^3$)–C($sp^3$) bond in **1b-2H** [1.709(2) Å] is much longer than that of previously reported analogs without a sulfur bridge[48,49]. Notably, even without the spiro ring at the elongated bond, which was assumed to be the key structural element causing the bond elongation, the bond length is comparable to the highest value hitherto reported for spiro-type dihydrophenanthrene derivatives [1.705(4) Å][50].

**Isolation of the "missing" structure, *o*-diphenoquinoid form C**

In addition to σ-bond form **B**, we anticipated that another structure, in which two diarylmethylene units are generated in the bay region by recombination of π-bonds following homolysis of a σ-bond, could also be a stable dicationic structure, i.e., tetraarylated *o*-diphenoquinoid form **C**. Indeed, DFT calculations at the CAM-B3LYP-D3/6-31G* level showed that *o*-diphenoquinoid form **C** was also obtained as an energy-minimized structure for the *ortho*-substituted dications **1$^{2+}$** (Figs. S27–S29). When the steric repulsion of the *ortho* substituents increases, the relative energy gap between the most stable *p*-quinoid form **A** and metastable *o*-diphenoquinoid form **C** diminishes considerably. Thus, dication **1c$^{2+}$** with *ortho* methyl groups has a greater chance than **1a$^{2+}$** with *ortho* fluorine atoms to adopt the form **C** structure (Table S13). To observe *o*-diphenoquinoid form **C**, dication salt **1c$^{2+}$[SbCl$_6$$^-$]$_2$** was prepared in 100% yield by treatment of **1c** with

two equivalents of magic blue (Fig. 5a). We tried to prepare single crystals of the **1c$^{2+}$** salt under several conditions but could not obtain suitable crystals, not even for salts with different counterions.

Therefore, based on the findings from the investigation of the hydride adduct of **1b$^{2+}$**, we aimed to convert **1c$^{2+}$** into a non-charged species and analyze its structure. Treatment of dication **1c$^{2+}$** with NaBH$_4$ gave target hydride adduct **1c-2H** as a deep-violet solid in 38% yield, together with (Ar$_4$QD)$_2$S **1c**, which was obtained in 60% yield via the 2e reduction of dication **1c$^{2+}$** (Fig. 5a). When the absorption spectrum was measured in CH$_2$Cl$_2$, neutral adduct **1c-2H** exhibited low-energy absorption bands in the visible region [$\lambda_{max}$/nm (log $\varepsilon$): 583 (4.17)], which indicates the presence of an effective π-conjugated structure. This result stands in contrast to the other neutral species, (Ar$_4$QD)$_2$S **1** and hydride adduct **1b-2H**, which do not exhibit such long-wavelength absorptions (Figs. 5b, S18). Finally, a single-crystal X-ray diffraction analysis unveiled that **1c-2H** adopts the *o*-diphenoquinoid form **C** structure, in which two exomethylenes are present on the same side of the molecule. The bond alternation of **1c-2H** is remarkable and strongly supports the assumption that **1c-2H** has an *ortho*-diphenoquinoidal character (Fig. 5c), which represents a "missing" link in closed-shell tetraaryldiphenoquinoidimethanes.

Moreover, even at higher temperatures in DMSO-$d_6$, sharp NMR signals were observed, which suggests that the thermally excited triplet species was not generated for **1c-2H**, even if there would be a contribution from the open-shell form (Fig. S21). The CASSCF

calculations also indicated that the biradical character is small in **1c-2H** as well as $(Ar_4QD)_2S$ **1** (Fig. S48), which is in good agreement with the experimental results. For **1c-2H**, the red-shifted absorption band extending to 650 nm was assigned to the π-π* transition derived from the HOMO to the LUMO, both of which are widely delocalized in the *o*-diphenoquinoid structure, as supported by TD-DFT calculations at the CAM-B3LYP-D3/6-31 G* level (Figs. S43, S47). The calculation results also show that a similar low-energy transition is absent in the other hydride adducts with *p*-quinoid form **A** or σ-bond form **B** geometries, thus revealing that a narrow HOMO-LUMO gap is characteristic of the geometry of the *o*-diphenoquinoid form **C** (Fig. S46). The pronounced steric repulsion caused by the methyl groups at the 2-position of aryl groups makes the formation of σ-bond geometries unfavorable, and thus, the formation of the *ortho*-diphenoquinoidal geometry was achieved due to the nature of the covalently connected QD framework. Accordingly, we have demonstrated that a redox-mediated strategy combined with control of the proximity of two electrophores is effective for the construction of diverse geometric structures from common $(Ar_4QD)_2S$ motifs that have two exomethylene units bridged directly in close proximity.

To elucidate the redox properties of **1c-2H**, electrochemical measurements were performed in $CH_2Cl_2$ (Fig. 5d). For **1c-2H**, the voltammogram showed that reversible two-stage 1e oxidation processes occur at $E_{1/2}^{ox1} = +0.65\,V$ vs. SCE and $E_{1/2}^{ox2} = +0.79\,V$, which differs from the apparent 2e redox process typical of arylated QD electrophores. Notably, these redox potentials are more cathodically shifted than those of the $(Ar_4QD)_2S$ **1c**, suggesting that **1c-2H**, obtained via a skeletal reorganization, shows superior electron-donating abilities due to the effective delocalization of the HOMO over the entire *o*-diphenoquinoid skeleton. To investigate the details of the redox behavior of *o*-diphenoquinoidimethane, an oxidative titration experiment, which was monitored using UV−vis−NIR spectroscopy was conducted on **1c-2H** using magic blue in $CH_2Cl_2$ (Fig. 5e, f). Upon addition of several aliquots of magic blue, sequential and drastic spectral changes were observed. First, vis−NIR bands at 689 and 1150 nm grew with an isosbestic point at 551 nm (**1c-2H** to **1c-2H$^{+}$**), and then strong absorptions at 574 nm and 764 nm gradually grew with an isosbestic point at 917 nm (**1c-2H$^{+}$** to **1c-2H$^{2+}$**). TD-DFT calculations suggested that the absorption bands of each redox state can be reasonably explained based on simulations of the corresponding optimized structures (Figs. S44, S47). These results demonstrate that the *o*-diphenoquinoid form, a "missing" diphenoquinoid isomer, can function as an NIR-switching material with multi-stage redox behavior.

In conclusion, we have designed and synthesized $(Ar_4QD)_2S$ **1**, in which the redox-active QD units are fused by a thiophene moiety. The two QD units are directly connected whilst another bridge formed by a sulfur atom suppresses the rotation of the two units to induce severe steric hindrance between the aryl groups, thus allowing to modulate the steric repulsion and/or electronic interaction between the two electrophores in specific redox states. We have unequivocally determined the structures of these redox states using spectroscopic, voltammetric, and X-ray diffraction techniques, which revealed that the most stable structure of the dicationic state can be precisely controlled by modulating the steric bulk of the substituents on the aryl groups in these molecules[36]. Indeed, the *p*-quinoid form **A** is preferred in dication **1a$^{2+}$**, which contains fluorine atoms as *ortho* substituents, while dication **1b$^{2+}$** with hydrogen *ortho* substituents adopts the σ-bond form **B** with an elongated C($sp^3$)−C($sp^3$) bond. Furthermore, a single-crystal X-ray diffraction analysis unveiled that the hydride adduct of **1c$^{2+}$**, **1c-2H**, with bulky methyl *ortho* substituents adopts the *o*-diphenoquinoid form **C**, which is an isomeric form of the diphenoquinoid species. *o*-Diphenoquinoid form **C** can be considered to be a "missing" link in the series of diphenoquinoid isomers and function as an NIR-switching redox system. We have demonstrated here that the appropriate control of the proximity of the two *p*-quinoid units in **1** can lead to, via the formation of π- or σ-bonds between the two electrophores, three different π-conjugated systems with distinct electronic properties. Thus, this study offers the successful modulation/control of different pathways toward electronically and/or thermodynamically stable structures in distorted multi-electrophore systems with overcrowded ethylenes. In addition to this molecular design guideline, incorporating other factors such as a 2D/3D structure or topology of the π-conjugation and electronic effects of substituents could lead to π-electron compounds with more tunable structural preferences and spin properties.

## Methods

All reactions were carried out under an argon atmosphere. All commercially available compounds were used without further purification. Dry MeCN was obtained by distillation from $CaH_2$ prior to use. Column chromatography was performed on silica gel 60N (KANTO KAGAKU, spherical neutral) of particle size 40–50 μm or Wakogel® 60N (neutral) of particle size 38–100 μm. $^1H$ and $^{13}C$ NMR spectra were recorded on a BRUKER Ascend™ 400 ($^1H$/400 MHz and $^{13}C$/100 MHz) spectrometer at 296 K unless otherwise indicated. IR spectra were measured on a Shimadzu IRAffinity-1S spectrophotometer using the attenuated total reflection (ATR) mode. Mass spectra were recorded on a JMS-T100GCV spectrometer in FD mode or a Q Exactive Plus in ESI positive mode by Dr. Eri Fukushi and Mr. Yusuke Takata (GS-MS & NMR Laboratory, Research Faculty of Agriculture, Hokkaido University). Elemental analyses were performed on an EXETER ANALYTICAL CE440 at the Center for Instrumental Analysis of Hokkaido University. Melting points were measured on a Yamato MP-21 and are uncorrected. UV−vis −NIR spectra were recorded on a JASCO V−770 spectrophotometer. Redox potentials ($E^{ox}$ and $E^{red}$) were measured on a BAS ALS-612EX by cyclic voltammetry and by differential pulse voltammetry in dry $CH_2Cl_2$ containing 0.1 M $Bu_4NBF_4$ as a supporting electrolyte. All of the values shown in the text are in $E$/V vs. SCE measured at the scan rate of 100 mV s$^{-1}$. Pt disk and wire were used as the working and counter electrodes, respectively. The working electrode was polished using a water suspension of aluminum oxide (0.05 μm) before use. DFT calculations were performed with the Gaussian 16W program package[51]. The geometries of the compounds were optimized by using the CAM-B3LYP-D3 method in combination with the 6−31G* basis set unless otherwise indicated. CASSCF calculations were performed with ORCA 6.0.1[52] using the 6−31G* basis set. A suitable crystal was selected and measured on a Rigaku XtaLAB Synergy (Cu-Kα radiation, $\lambda = 1.54184\,Å$) with HyPix diffractometer. Using Olex2[53], the structure was solved with the SHELXT[54] structure solution program using Intrinsic Phasing and refined with the SHELXL[55] refinement package using Least Squares minimization.

## Data availability

The X-ray crystallographic coordinates for structures reported in this study have been deposited at the Cambridge Crystallographic Data Centre (CCDC), under deposition numbers 2342072−2342077, 2342079−2342085, and 2400963. These data can be obtained free of charge from The Cambridge Crystallographic Data Centre via www. ccdc.cam.ac.uk/data_request/cif. All data are available from the corresponding author upon request. Source Data are provided with this manuscript. Source data are provided with this paper.

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

## Acknowledgements
This work was supported by Grant-in-Aid from MEXT and JSPS (Nos. JP24K23093 and JP25K18023 to T.H., JP25K08604 to T.S., and JP23H04011, JP25H00873, and JP25H01259 to Y.I.) and JST PRESTO (JPMJPR23Q1 to Y.I.). T.H. is grateful for TOBE MAKI Scholarship Foundation. This work was also supported by the Research Program of "Five-star Alliance" in "NJRC Mater. & Dev." MEXT.

## Author contributions
T.H. and Y.I. developed the concept of this study. T.H. conducted the synthetic and spectroscopic experiments as well as the theoretical calculations. M.K. conducted an oxidative experiment of **1c-2H**. Y.I. and T.S. supervised the project. T.H. and Y.I. prepared the manuscript with feedback from all authors.

## Competing interests
The authors declare no competing interests.
