## [Transparent Peer Review file · Nature Communications]

Diverse Redox-Mediated Transformations to Realize the *Para*-Quinoid, σ -Bond, and *Ortho*-Diphenoquinoid Forms

Corresponding Author: Professor Yusuke Ishigaki

Version 0:

Reviewer comments:

Reviewer #1

(Remarks to the Author)

Please, see the attached PDF file, "Comments for Authors".

Reviewer #2

(Remarks to the Author)

In this manuscript, the authors have synthesized three derivatives of sulfur-bridged quinodimethane 1a, 1b, and 1c, each incorporating different aryl substituents. The *p*-quinoid, σ -bond, and unusual *o*-quinoid forms were observed by modulating the steric hindrance of the aryl groups. Cationic species 1c-2H²⁺ exhibit near-infrared (NIR) absorption extending up to 1400 nm, thereby demonstrating their potential applications in NIR-switching materials. Overall, this study demonstrates high scientific quality. This reviewer recommends some revisions before it is considered for publication.

1. For the high-resolution mass spectrometry (HRMS) data provided in the Supporting Information, the measurement errors should be specified for all compounds.
2. In the reduction of 1b²⁺ and 1c²⁺, why was 1b-2H obtained quantitatively, whereas only 38% of 1c-2H was recovered?
3. On Page 9, the statement "Moreover, even at higher temperatures in DMSO-d₆, sharp NMR signals were observed, which suggests that the open-shell character in 1c-2H is negligible" should be revised. Sharp peaks at elevated temperatures are indicative of a low contribution from the triplet state, rather than a negligible open-shell character.
4. The aromaticity of the dinaphthothiophene (DNT) in different forms could be further elucidated by analyzing the ACID plots and NICS values for the three different forms of 1a²⁺, 1b²⁺, and 1c²⁺.
5. Since quinoidal structures are typically associated with open-shell diradical forms, what are the ground states and diradical characters of 1a, 1b, and 1c?

Reviewer #3

(Remarks to the Author)

General comments,

In this paper, the authors designed and synthesized a series compound of (Ar₄QD)₂S, in which the redox-active QD units are fused by a thiophene moiety. The two QD units are directly connected, while another bridge was formed by a sulfur atom suppresses the rotation of two units to induce severe steric hindrance between the aryl groups. The configuration allows to modulate the steric repulsion and/or electronic interaction between the two electrophores in specific redox states. The authors have unequivocally determined the structures of these redox states using spectroscopic, voltametric, and X-ray diffraction methods, which revealed that the most stable structure of dicationic state can be precisely controlled by modulating the steric bulk of the substituents on the aryl groups in these molecules. However, the reviewer has some concerns, especially missing completed discussions on the electrochemical transformations as claims by the authors, which should be addressed before this paper can be a potential publication on Nature Communications.

Specific comments,

1. The author claimed that the UV-Vis spectra of 1c-2H stands in contrast to the other neutral species, (Ar4QD)2S 1 and hydride adduct 1b-2H, which do not exhibit such long-wavelength absorbance. It is suggested to further compare the UV-Vis spectra of 1a²⁺ and 1c²⁺, and provide the corresponding discussions. It is obvious that 1a and 1c have more similar open-shell structures. Additionally, the authors should explain the shift in the UV-Vis absorbance bands, such as the blue shift of 1b²⁺.
2. The redox properties of 1a and 1c-2H were characterized by CV curves. However, the CV data of 1b-2H and its explanation are missing.
3. It is claimed that AQD I-a is unlike 1a, a different structural preference for a folded geometry in the neutral state and a twisted geometry in the dication state. What is the neutral state and twisted geometry of ADQ I-a?
4. The authors used diverse electrochemical transformations to realizes the Para-Quinoid, σ -Bond, and Ortho-Diphenoquinoid forms. However, the electrochemical experiments are merely the CV data for presenting the redox properties of different products. How did electrochemical transformations emerge?
5. There are significant data in the SI appendix. The sequence of text does not align well with the order of the images of SI appendix. For instance, in the main text, the authors mentioned the data of S41. While it is difficult for the readers to find such a data. It is suggested to carefully arrange the data in main text and SI appendix.

Version 1:

Reviewer comments:

Reviewer #1

(Remarks to the Author)

The changes made by the authors in the manuscript are almost fully adequate and the clarity of the paper is improved, especially by adding explanation of the effect of steric distortion on the co-planarity of the participating π -subsystems and the subsequent effect on the HOMO energy level and thus, oxidation potential.

As final and crucial suggestions, we state the following:

- Page 9, last paragraph: the statement about the 1c-2H compound having the negligibly small open-shell character is not rigorously correct. In the presented set of compounds, for which diradical characters were calculated from the CASSCF results, all of them are around $y_0 = 0.1$ and higher. This is an important degree of open-shell character for the reactivity. It is more appropriate to state that open-shell character is small, but non-negligible in terms of chemical reactivity, even if this character is not detectable with some of the experimental methods. If authors need to categorize the degree of open-shell character they can pre-define what small and medium diradical characters mean in the context. Please correct all the instances of such use of the term, because for the readers in this field, such wording will cause confusion.
- Page 11, end of Conclusions: "geometry of the π -conjugation" is an ill-defined phrase. More appropriate to use: "topology of the π -conjugation"/"topology of the π -conjugated system"/"geometry of the π -conjugated system". The preferred term when one refers to the graph relation between overlapping/crossing/parallel π -chains is "topology" rather than geometry. This is because one can achieve equivalent topologies of the π -systems with different geometries, but two different topologies of the π -system cannot be achieved by the same geometry, by definition.

Reviewer #2

(Remarks to the Author)

The authors have addressed all of my questions. I recommend the acceptance of this manuscript for publication.

Reviewer #3

(Remarks to the Author)

The authors have addressed the raised concerns. This work can be a potential publication on Nature Communications.

<Addressing the comments of Reviewer-1>

Reviewer #1 (Remarks to the Author):

Please, see the attached PDF file, "Comments for Authors".

Recommendation: This manuscript is adequate for the publication, but should be revised by authors based on reviewer feedback and modified version should be reviewed again before the manuscript is finally accepted.

Important remarks and suggestions:

The article by Harimoto et al. exemplifies how co-dependent electronic and molecular structure are tuned by the presence of steric hindrance.

By increasing the steric hindrance from hydrogen, to fluorine, to methyl attached to the aryl groups of diarylmethylene terminal units of sulfur bridged quinodimethane derivative, the multiconfigurational electronic structure is tuned.

The ground state electronic wavefunction of presented compounds (**1a**, **1b**, **1c**) is the combination of multiple valence bond forms (VBFs, i.e. resonance structures) of the π -conjugated subsystem. The relative contribution of each VBF depends on the specific build of the molecule as changing some groups would induce the compound to assume significantly different geometry. The relative contribution of each VBF in the compounds is implied by authors based on the analysis of the geometry given by x-ray crystallography and is further corroborated by the electrochemical behavior of **1a**, **1b**, **1c** and structures/properties of their dication species.

The article is well presented, accessible to chemists from broad backgrounds and is interesting for the researchers in various subfields of molecular chemical science. Nevertheless, the article needs a revision to make some improvements in insights, fill the gaps in discussion and correct a few technical points.

=> We sincerely appreciate the reviewer's valuable comments and recommendation. The order of the Experimental Section and the Theoretical Study in the Supporting Information was changed to make it easier for readers to understand. Based on the suggestions, the manuscript was revised, and the details are described below.

- The quality of the three-dimensional drawings of the crystal structure should be improved as sometimes the relevant structural details that would lead to the specific features of the electronic structure are elusive.

=> We tried to make the angle of view of the ORTEP drawings reflect the 3D structure as much as possible, however, it was difficult to further improve the quality of the 3D drawings of the X-ray crystal structures because the molecules are sterically distorted with multiple aryl groups. We have uploaded the original CIF files for each molecule so that readers can see the detailed structure of each molecule.

Furthermore, the authors need to discuss the effect of dihedral angles between fragments of the molecule more rigorously. For example, the π -conjugated diarylmethylene fragments are not fully coplanar with the π -conjugated core of molecules (Tables S3-S5) and thus, the overlap between p orbitals at the interface of different fragments is not optimal for π -bonding. This obviously affects the electronic

structure. Therefore, this analysis should be integrated into the main text concisely but more explicitly.

=> As pointed out by the reviewer, the diarylmethylene moiety of the neutral donors adopts distorted folded conformations due to steric repulsion between the aryl groups and the central skeleton, causing the π -conjugated core to deviate from the co-planar plane. These folded structures are common to a series of benzo-fused quinodimethanes. The HOMOs of the neutral donor **1a-1c** are mainly distributed in the diarylmethylene moieties, and the HOMO levels are affected by both steric and electronic effects of the *ortho* substituent. Since the steric repulsion of the *ortho* substituent is more pronounced in **1a** with F atoms and **1c** with Me groups than in **1b** with H atoms, the dihedral angles β around the aryl group and methylene moiety become larger, resulting in lower co-planarity (β_{ave}^{calc} **1a**: 52.5° and **1c**: 52.6° vs **1a**: 39.6°). The co-planarity is effective for the π conjugation between the methylene moiety and the methoxyphenyl groups, resulting in the elevated HOMO level in **1b** (Fig. S41). This is consistent with the results of the CV measurements (Fig.S16), which showed that **1b** was most easily oxidized due to the highest HOMO level among **1a-1c** (E_{peak}^{ox1} : +0.73 V vs SCE). Neutral donor **1c** has weak electron-donating Me groups at the *ortho*-position on the aryl group, which could electronically contribute to the increase in the HOMO level, but the conjugation between the aryl groups and the exomethylene moieties is not very effective due to the lower co-planarity, resulting in the slightly more positive oxidation potential (E_{peak}^{ox1} : +0.85 V vs SCE for **1c**) than that for **1b**. Compared to these two derivatives **1b** and **1c**, in the case of fluorinated donor **1a**, both the steric and the electronic effects of electron-withdrawing F atoms are dominant, inducing a significant decrease in the HOMO level, and thus the oxidation potential was observed in the most anodic region (E_{peak}^{ox1} : +1.00 V vs SCE). In summary, it is suggested that the oxidation potentials (i.e., HOMO levels) of the neutral donor are affected by both steric and electronic effects of *ortho* substituents, which is in common with our previous study (*Chem. Eur. J.* **2023**, *29*, e202203899 [DOI: 10.1002/chem.202203899]) for the reference monomeric AQDs **I**. Since the effects of *ortho* substituents on the frontier orbitals of neutral AQDs have already been discussed in our previous study, a detailed discussion on these (Ar₄QD)₂S species is added to the Supporting Information (p.S31), as follows:.

"The HOMOs of the neutral donor **1a-1c** are mainly distributed in the diarylmethylene moieties, and the HOMO levels are affected by both steric and electronic effects of the *ortho* substituent. Since the steric repulsion of the *ortho* substituent is more pronounced in **1a** with F atoms and **1c** with Me groups than in **1b** with H atoms, the dihedral angles β around the aryl group and methylene moiety become larger, resulting in lower co-planarity (β_{ave}^{calc} **1a**: 52.5° and **1c**: 52.6° vs **1a**: 39.6°). The co-planarity is effective for the π conjugation between the methylene moiety and the methoxyphenyl groups, resulting in the elevated HOMO level in **1b** (Fig. S41). This is

consistent with the results of the CV measurements, which showed that **1b** was most easily oxidized due to the highest HOMO level among **1a-1c** ($E_{\text{peak}}^{\text{ox1}}$: +0.73 V vs SCE). Neutral donor **1c** has weak electron-donating Me groups at the *ortho*-position on the aryl group, which could electronically contribute to the increase in the HOMO level, but the conjugation between the aryl groups and the exomethylene moieties is not very effective due to the lower co-planarity, resulting in the slightly more positive oxidation potential ($E_{\text{peak}}^{\text{ox1}}$: +0.85 V vs SCE for **1c**) than that for **1b**. Compared to these two derivatives **1b** and **1c**, in the case of fluorinated donor **1a**, both the steric and the electronic effects of electron-withdrawing F atoms are dominant, inducing a significant decrease in the HOMO level, and thus the oxidation potential was observed in the most anodic region ($E_{\text{peak}}^{\text{ox1}}$: +1.00 V vs SCE). In summary, it is suggested that the oxidation potentials (i.e., HOMO levels) of the neutral donor are affected by both steric and electronic effects of *ortho* substituents, which is in common with our previous study (*Chem. Eur. J.* **2023**, *29*, e202203899 [DOI: 10.1002/chem.202203899]) for the reference monomeric AQDs I."

- Electronic structure problem should be more precisely defined for the explored species. Hence, rigorous theoretical analysis based on resonance theory/valence bond theory within a multiconfigurational quantum chemistry formalism serves useful. Also, the description of the electronic structure by DFT methods is rather simplistic and insufficient for the appropriate theoretical and computational analysis.

=>First of all, both neutral and dicationic species are all closed-shell compounds, which do not exhibit diradical character even at high temperatures. Actually, as pointed out by the reviewer, some quinodimethane (QD)-based systems exhibit open-shell character based on the resonance structures (e.g., *Acc. Chem. Res.* **2017**, *50*, 977–987 [DOI: 10.1021/acs.accounts.7b00004], *J. Am. Chem. Soc.* **2020**, *142*, 5408-5418 [DOI: 10.1021/jacs.0c01003], and *Angew. Chem. Int. Ed.* **2022**, *61*, e 202205729 [10.1002/anie.202205729]). However, our systems, such as dibenzo-annulated *para*-quinodimethane, anthraquinodimethane (AQD), derivatives with bulky diarylmethylene units and bisquinodimethane (BQD) analogues, generally do not exhibit open-shell character. Due to the large substituents on the exomethylene units, AQD and BQD derivatives adopt highly strained folded conformations with closed-shell characters (our previous work: *Chem. Commun.* **2021**, *57*, 7201-7214 [DOI: 10.1039/D1CC02260A], *J. Am. Chem. Soc.* **2021**, *143*, 3306-3311 [DOI: 10.1021/jacs.1c00189], *Chem. Eur. J.* **2023**, *29*, e202203899 [DOI: 10.1002/chem.202203899], and *Chem. Eur. J.* **2023**, *29*, e202301476 [DOI: 10.1002/chem.202301476]). To make the molecules with open-shell characters, it needs specific molecular design, e.g., oligoanthraquinodimethane derivatives (*J. Am. Chem. Soc.* **2012**, *134*, 14513-14525 [DOI: 10.1021/ja3050579], *J. Am. Chem. Soc.* **2022**, *144*, 7479-7488 [DOI: 10.1021/jacs.2c02318], and *J. Am. Chem. Soc.* **2023**, *145*, 2596-2608 [DOI: 10.1021/jacs.2c12574]) or more flexible AQDs (*Angew. Chem. Int. Ed.* **2020**, *59*, 6581-6584 [DOI: 10.1002/anie.201916089] and *Chem. Eur. J.* **2024**, *30*, e202400916 [DOI: 10.1002/chem.202400916]). For these QD-based systems, the energy gap between singlet and triplet ($\Delta E_{\text{S-T}}$) is, in general, very small, and thus

thermally excited triplet species can be easily generated under usual conditions. The ΔE_{S-T} and the contribution of such triplet species can be estimated by variable-temperature (VT) ^1H NMR experiments in combination with theoretical calculations.

As pointed out by the reviewer, it is expected that the CASSCF calculation, including the multiconfigurational quantum chemistry formalism, will provide detailed information on the electronic structure, such as open-shell nature. The details are shown below.

Firstly, it is crucial to study the simplified model of the presented compound set, which would reveal the electronic structure based on the topology of π -conjugation.

One can use the methylene linker to avoid steric hindrance and maintain the same topology of π -conjugation. This is why the model system below is important to analyze.

Note that π -bonds can be rearranged from the drawn, but the overall topology of π -conjugation is defined by tailoring the geometry.

=> To avoid steric hindrance and maintain the same topology of π -conjugation, the methylene-bridged molecule that the reviewer suggested would be an important reference system. However, when an optimization calculation was performed for the proposed molecular structure, the angular distortion of the seven-membered ring caused the main skeleton to be distorted into an undesired U-shaped structure. Therefore, we employed a molecule with H atoms instead of aryl groups as the least distorted reference structure, which adopts a folded structure similar to that of the octaarylated neutral donors **1a-1c**.

The electronic structure of this model system should be analyzed as a reference for the compounds **1a**, **1b**, **1c**, to better understand how these different groups modulate the properties initially determined by the topology of π -conjugation in the case of negligible steric hindrance. This will improve the insight into the rational design principles. Secondly, DFT description of such multiconfigurational compound is usually incomplete since, in addition to being single-reference method, DFT shows significant dependence of energy gaps and properties on the choice of exchange-correlation functional. In order to better understand the electronic structure of this multiconfigurational systems, it is highly recommended to use multireference methods such as complete active space self-consistent field (CASSCF) or at

least complete active space configuration interaction (CASCI) with proper guess orbital set. The active space for this electronic structure problem is not too large and is computationally tractable with common computing clusters (probably CAS(14,14) is the sufficient size of active space and moderate-sized basis set is enough), as aryl groups in diarylmethylene parts do not participate in the subspace of frontier π -orbitals. The authors can use DFT-optimized geometries and for each geometry (different DFT- optimized forms) for each compound, determine the excited states by CASSCF/CASCI method. The results can be analyzed in terms of CASSCF natural orbitals and their occupation numbers to obtain the description of electron density distribution of most relevant part of orbital set for a given electronic problem.

=> By following the reviewer's suggestion, CASSCF calculations using ORCA 6.0.1 were performed for the singlet states of neutral donors **1a-1c** and several reference molecules. The calculation procedure is as follows: 1) Perform the structural optimizations at the CAM-B3LYP-D3/6-31G(d) level. 2) Calculate unrestricted natural orbitals and quasi-restricted orbitals based on the optimized structures and examine the π -bonding orbitals distributed mainly in the parent skeletons and exomethylene moieties that can contribute to the open-shell character, which are reflected in the active space. 3) Conduct the CASSCF calculations (CAS(14,14)) at the 6-31G(d) level to analyze the occupation number for the LUMO (LUNO) and estimate the diradical index y_0 (e.g., *Phys. Chem. Chem. Phys.* **2014**, *16*, 9565-9571 [DOI: 10.1039/C4CP00939H], *Chem. Sci.* **2018**, *9*, 6107-6117 [DOI: 10.1039/C8SC01999A], *J. Org. Chem.* **2023**, *88*, 8553–8562 [DOI: 10.1021/acs.joc.3c00482]). The molecular structures and their y_0 values are summarized below and added to the Supporting Information (Figure S48, p. S77).

CASSCF(14,14) 6-31G* calculations of optimized structures at the CAM-B3LYP-D3/6-31G* level

(1) Neutral donors **1a-1c**

The diradical character y_0 of the experimentally isolated neutral donors **1a-1c** is about 0.104~0.124. There is no significant difference regardless of the aryl group, suggesting a closed-shell character, which is consistent with the experimental results. Comparison of reference molecules **1H** and **1H''**, which have less steric distortion, suggests that the

presence or absence of a benzo-fused ring, which is crucial for the Clar's aromatic π -sextet rule discussed below, has a significant effect on the y_0 value. On the other hand, comparing **1b** with aryl groups (4-methoxyphenyl groups) and its reference molecule **1b''**, there is almost no difference in the y_0 value, which is much smaller than those of **1H** and **1H''**, suggesting that the steric hindrance and distortion caused by the introduction of aryl groups significantly reduce the diradical character, irrespective to the presence or absence of a benzo-fused ring. In any case, since even **1H''** ($y_0=0.143$) has a smaller value than Chichibabin's Hydrocarbon ($y_0=0.296$), which shows some contributions of open-shell character, we can conclude that, due to the distortion of the diaryl-substituted exomethylene moiety and the fusion of thiophene ring, the open-shell character is almost negligible in these systems.

(2) **1c-2H** with *o*-diphenoquinoid form

The diradical character y_0 of the experimentally isolated neutral *o*-diphenoquinoid **1c-2H** is about 0.123, suggesting that there is less contribution of open-shell character. Indeed, the experimental results indicate that **1c-2H** exhibits a closed-shell character. When calculated for the reference molecule **1H-2H** without aryl groups, its y_0 value was slightly higher than that of **1-2H**. This is presumably due to the reduced steric distortion (increased co-planarity) of the central skeleton. Furthermore, when **1H-2H''** and **1-2H''** without benzo-fused rings were calculated, the y_0 values were significantly larger, suggesting that the presence of benzo-fused rings in the *o*-diphenoquinoid form is an important factor for the decrease in the open-shell characters.

Therefore, these experimental and theoretical results suggest that the neutral donor **1a-1c** and **1c-2H** isolated in this study are closed-shell species, which is common to previously reported benzo-fused quinodimethanes (our previous work: *Chem. Commun.* **2021**, 57, 7201-7214 [DOI: 10.1039/D1CC02260A], *J. Am. Chem. Soc.* **2021**, 143, 3306-3311 [DOI: 10.1021/jacs.1c00189], *Chem. Eur. J.* **2023**, 29, e202203899 [DOI: 10.1002/chem.202203899], and *Chem. Eur. J.* **2023**, 29, e202301476 [DOI: 10.1002/chem.202301476]). It is noted that, in terms of redox-mediated diversification of molecular structures, the open-shell character of neutral species **1a-1c** does not affect the purpose and results of this paper.

In addition, the contents of the Table S1 and Table S2, describing the different reference system (**1a'-1c'** and **1a''-1c''**, and **1c-2H''**) should be more carefully discussed and may have to be re-formulated and re-evaluated. The important difference between **1a''**, **1b''**, **1c''**, **1c-2H''** (second set of chosen reference systems) and **1a**, **1b**, **1c**, (primary systems) is that compounds **1a''** to **1c''** and **1c-2H''** do not have terminal fused benzene rings as opposed to **1a**, **1b**, **1c**, that significantly affects the electronic structure. That is why the diradical character would much be higher in the case of **1c-2H''** compared to **1c-2H** because shifting from closed-shell to open-shell form allows to gain two Clar's π -sextets for **1c-2H''**, while there is no such change for **1c-2H** (also see comment 2 below).

=> Consistent with the reviewer's keen insight, CASSCF calculations suggest that the diradical character of **1H-2H''** and **1-2H''** without benzo-fused rings is increased compared to that of **1H-2H** and **1-2H** with benzo-fused rings. Thus, it is suggested that the benzo-fusion in the *o*-diphenoquinoid form is an important factor for the decrease in the open-shell characters (vide supra). This is also consistent with a qualitative explanation for the acquisition of the Clar's aromatic π -sextet in the diradical state.

Note that increased steric hindrance between aryl groups in the presented systems would distort the internal six-membered ring (Tables S3-S10) so that the aromatic resonance energy would be noticeably diminished compared to benzene and terminal benzenoid rings.

=> As expected, CASSCF calculations for *o*-diphenoquinoid forms showed an increase in the diradical character of the reference molecules **1H-2H''** and **1H-2H** without aryl groups compared to the arylated **1-2H''** and **1-2H**. This can be explained by the enhanced π -conjugated co-planarity of the main skeleton due to reduced steric repulsion induced by the absence of the aryl groups.

The statement that **1c-2H** is singlet diradical (row 2 from below and column 5 of Table S2) seems to contradict the statement in the main text, page 9 (also see comment 2). Authors must correct **1c-2H** by **1c-2H''** (second last row of Table S2) as this error causes confusion in the reader.

=> We would like to thank the reviewer for the careful reading. Amended as suggested.

Also, it is not fully clear what authors mean by singlet diradical and which criteria are applied for such categorization. As mentioned, the analysis of CASSCF and/or CASCI results would give us more detailed and qualitatively accurate description. This is necessary to determine the degree to which the electronic structure of the given state is singlet diradical. Therefore, analysis of diradical character based on the frontier natural orbital occupation numbers and description of excited states by CASCI/CASSCF in such cases is crucial.

=> CASSCF calculations were performed to estimate natural orbital coefficients and evaluate the diradical character. As a result, all the derivatives isolated in this study are estimated to be singlet closed-shell species. For general π -conjugated systems that do not involve a dynamic change in molecular structure, the open-shell character can be discussed qualitatively on the basis of resonance theory. On the other hand, in highly distorted π -conjugated systems, as in this study, the structures of the singlet closed-shell state and the singlet open-shell state can change dynamically. Especially in arylated QD systems, the structures of the singlet closed-shell (folded structure) and the singlet diradical (twisted structure) are thoroughly different (e.g., *Angew. Chem. Int. Ed.* **2024**, 63, e202316753 [DOI: 10.1002/anie.202316753] and *Chem. Eur. J.* **2024**, 30, e202400916 [DOI: 10.1002/chem.202400916]). Therefore, the local-minimized structures and relative energies of each state can be obtained by conventional DFT calculations. In this paper, we treat systems that induce dynamic changes in conformation, and thus we distinguish the terms "singlet closed-shell states" and "singlet open-shell states".

Comments:

1. Figure 1. The electronic structure of forms A, B, C may not be clear to the general reader: Is it a diradical, dication or closed-shell? It is highly desirable that this is visually better clarified in the chemical drawings of compounds.

=> Both neutral and dicationic species are all closed-shell compounds, which is supported by experimental results and theoretical calculations. In Figure 1, three forms **A-C** are represented as possible structures that can be adopted in these systems, and their open-shell character is not mentioned at this point (since all isolated species show closed-shell character). In addition, since Figure 1 shows a general schematic illustration for three forms **A-C** including not only dicationic states but also neutral states (hydride adducts), it is desirable to describe the structures comprehensively in Figure 1. However, to avoid readers' misleading, we amended the main text that $(Ar_4QD)_2S$ systems are closed-shell donors as follows:

"Based on these considerations, we designed and synthesized sulfur-bridged closed-shell π -electron systems **1** with two extra benzene rings $[(Ar_4QD)_2S]$ (Fig. 1c)."

2. "Moreover, even at higher temperatures in DMSO-*d*6, sharp NMR signals were observed, which suggests that the open-shell character in **1c-2H** is negligible, even though in the open-shell form, resonance energy can be gained by aromatization of the two rings (Fig. S40). " – page 9.

If one analyzes the valence bond forms properly, one sees that this reasoning is based on inaccurate assumption. In the simplified representations of the compound **1c-2H** (this applies to **1a**, **1b** and **1c**, analogically too), the count of Clar's sextets remains the same irrespective of closed-shell or open-shell structure. This is exactly the reason why the open-shell character is negligible, because there is no significant resonance energy gain in open-shell forms (two Clar's sextets) compared to closed-shell form (two Clar's sextets). This must be addressed in the revised version of the article by more rigorous analysis suggested here. Moreover, as mentioned above, the six-membered rings bridged by sulfur are geometrically more distorted than the terminal six-membered rings. Hence, the aromatic resonance energy of the terminal six-membered rings is expected to be larger compared to internal six-membered rings upon assuming the Clar's π -sextet configuration. Since the closed-shell structure has Clar's π -sextets in the terminal rings, there is no significant gain (if any) in aromatic resonance energy upon opening the shell and thus, the open-shell character remains negligible. Therefore, the correction must be made in the main text and as mentioned above, more careful comparison between presented primary compounds and assumed reference compounds is necessary with the new reference compound that we suggested above.

=> As suggested by the reviewer, since there is no significant resonance energy gain in open-shell forms compared to closed-shell state, we amended the main text as follows. In

addition, based on the CASSCF calculations recommended by the reviewer, we added the explanation that **1c-2H** has negligible open-shell character.

“Moreover, even at higher temperatures in DMSO- d_6 , sharp NMR signals were observed, which suggests that the thermally excited triplet species was not generated for **1c-2H**, even if there would be a contribution from the open-shell form (Fig. S21). The CASSCF calculations also indicated that the biradical character is negligibly small in **1c-2H** as well as (Ar₄QD)₂S **1** (Fig. S48), which is in good agreement with the experimental results.”

3. “Thus, this study is the very first example of the successful modulation/control of different pathways toward electrochemically and/or thermodynamically stable structures in multi-electrophore systems. This work can be expected to lead to the development of a library of unprecedented π -electron compounds.” – page 11. Even though the modulation of assumed ground-state geometry as a function of steric hindrance is remarkable and is quite well explained, in order to generate the large library of important π -electron compounds, one needs to study the influence of other factors such as topology of π -conjugation and electronic effects (electron-donating/withdrawing). The modulation of structures only as a function of steric effects does not necessarily allow the desired degree of freedom in rational design. In π -conjugated compounds, the principal factor that controls the properties is the topology of the π -conjugation. One can state that steric effects are final fine-tuning of the co-dependent geometry and electronic structure of the π -conjugated compound, as the topology of the π -system sets the constraints and defines the nature of contributing valence bond forms in the electronic structure, contribution of which can then be modulated with adjusting the steric hindrance. Thus, it would be more appropriate to moderate the final sentence of the article with more specific statement, such as stating the mechanism of control of structures in multi-electrophore systems and the specific degree of freedom it allows in the presented π -conjugated systems, rather than ambiguously stating that this approach can be expected to lead to an unprecedented library of π -electron compounds. In other words, authors should specify the degree of variety you could expect in the library of π -electron compounds because based on the steric effects alone, one cannot generate a significant portion of chemical space of π -conjugated compounds with a sufficient variety of properties. Even though this does not invalidate the presented work, the concluding statement is exaggerated and should not go outside the scope of the article. The completed experimental work seems well-performed for the study of properties and provides very useful data for the analysis of controlling the preferred form of the sulfur-bridged quinodimethane derivatives.

=> According to the reviewer’s suggestion, we considered that a more precise statement would be desirable as a conclusion to this paper.

Therefore, the statement was amended as follows:

"Thus, this study is the very first example of the successful modulation/control of different pathways toward electronically and/or thermodynamically stable structures in distorted multi-electrophore systems with overcrowded ethylenes. ~~This work can be expected to lead to the development of a library of unprecedented π -electron compounds.~~ In addition to this molecular design guideline, incorporating other factors such as a geometry of the π -conjugation and electronic effects of substituents could lead to unprecedented π -electron compounds with more tunable structural preferences and spin properties."

To summarize, as a theoretical reviewer, I would still strongly recommend that one of the reviewers of this manuscript has expertise in the used experimental methods and experience in structural determination or synthesis of the presented or related class of compounds.

=> Based on the other two reviewers' suggestions and comments, we performed additional experimental and theoretical investigations. Our response is shown below.

Reviewer #2 (Remarks to the Author):

In this manuscript, the authors have synthesized three derivatives of sulfur-bridged quinodimethane **1a**, **1b**, and **1c**, each incorporating different aryl substituents. The p-quinoid, σ -bond, and unusual o-quinoid forms were observed by modulating the steric hindrance of the aryl groups. Cationic species **1c-2H**²⁺ exhibit near-infrared (NIR) absorption extending up to 1400 nm, thereby demonstrating their potential applications in NIR-switching materials. Overall, this study demonstrates high scientific quality. This reviewer recommends some revisions before it is considered for publication.

=> We are deeply grateful for the reviewer's valuable comments and recommendations. Based on the suggestions, the manuscript was revised, and the details are described below.

1. For the high-resolution mass spectrometry (HRMS) data provided in the Supporting Information, the measurement errors should be specified for all compounds.

=> Based on the reviewer's suggestion, Supporting Information was amended.

2. In the reduction of **1b**²⁺ and **1c**²⁺, why was **1b-2H** obtained quantitatively, whereas only 38% of **1c-2H** was recovered?

=> As described in the literature (e.g., *J. Am. Chem. Soc.* **1953**, *75*, 215-219 [DOI: 10.1021/ja01097a057] and *Catalysis Today* **2011**, *170*, 3-12 [DOI: 10.1021/ja01097a057]), NaBH₄ works not only as a hydride donor but also as an electron donor (reductant). In our systems, the reduction potential of **1c**²⁺ ($E_{\text{red}}^{\text{peak}} = +0.17$ V vs. SCE) is much less positive than that of **1b**²⁺ ($E_{\text{red}}^{\text{peak}} = +0.74$ V vs. SCE). As a result, **1c**²⁺ undergoes both the two-electron reduction and the addition of hydrides to give **1c** and **1c-2H** in 60% and 38% yields, respectively, while the treatment of **1b**²⁺ with NaBH₄ yielded **1b-2H** quantitatively. In addition, steric hindrance may affect both yields because the reaction center of **1c**²⁺ is more crowded than that of **1b**²⁺ due to the *ortho* substituents on the aryl groups. The important thing is that both reduction reactions proceed cleanly, and there were no other byproducts after the reduction reaction using NaBH₄.

3. On Page 9, the statement "Moreover, even at higher temperatures in DMSO-d₆, sharp NMR signals were observed, which suggests that the open-shell character in **1c-2H** is negligible" should be revised. Sharp peaks at elevated temperatures indicate a low contribution from the triplet state, rather than a negligible open-shell character.

=> As pointed out by the reviewer, some quinodimethane (QD)-based systems exhibit open-shell character based on the resonance structures (e.g., *Acc. Chem. Res.* **2017**, *50*, 977–987 [DOI: 10.1021/acs.accounts.7b00004], *J. Am. Chem. Soc.* **2020**, *142*, 5408-5418 [DOI: 10.1021/jacs.0c01003], and *Angew. Chem. Int. Ed.* **2022**, *61*, e 202205729 [10.1002/anie.202205729]). On the other hand, our systems, such as dibenzo-annulated *para*-quinodimethane, anthraquinodimethane (AQD), derivatives with bulky diarylmethylene units and bisquinodimethane (BQD) analogues, generally do not exhibit open-shell character. Due to the large substituents on the exomethylene units, AQD and BQD derivatives adopt

highly strained folded conformations with closed-shell characters (our previous work: *Chem. Commun.* **2021**, 57, 7201-7214 [DOI: 10.1039/D1CC02260A], *J. Am. Chem. Soc.* **2021**, 143, 3306-3311 [DOI: 10.1021/jacs.1c00189], *Chem. Eur. J.* **2023**, 29, e202203899 [DOI: 10.1002/chem.202203899], and *Chem. Eur. J.* **2023**, 29, e202301476 [DOI: 10.1002/chem.202301476]). To make the molecules with open-shell characters, it needs specific molecular design, e.g., oligoanthraquinodimethane derivatives (*J. Am. Chem. Soc.* **2012**, 134, 14513-14525 [DOI: 10.1021/ja3050579], *J. Am. Chem. Soc.* **2022**, 144, 7479-7488 [DOI: 10.1021/jacs.2c02318], and *J. Am. Chem. Soc.* **2023**, 145, 2596-2608 [DOI: 10.1021/jacs.2c12574]) or more flexible AQDs (*Angew. Chem. Int. Ed.* **2020**, 59, 6581-6584 [DOI: 10.1002/anie.201916089] and *Chem. Eur. J.* **2024**, 30, e202400916 [DOI: 10.1002/chem.202400916]). For these QD-based systems, the energy gap between singlet and triplet (ΔE_{S-T}) is, in general, very small, and thus thermally excited triplet species can be easily generated under usual conditions. The ΔE_{S-T} and the contribution of such triplet species can be estimated by variable-temperature (VT) ^1H NMR experiments in combination with theoretical calculations. Indeed, we carried out VT- ^1H NMR and theoretical studies for *ortho*-diphenoquinoid species, indicating that the open-shell property of **1c-2H** is negligibly small. Moreover, we newly performed CASSCF calculations for our systems as well as model compounds, as suggested by another reviewer. As a result, all compounds in this paper do not have diradical character, even for **1-2H** that has less steric strain.

Therefore, the statement was amended as follows:

"Moreover, even at higher temperatures in DMSO- d_6 , sharp NMR signals were observed, which suggests that the thermally excited triplet species was not generated for **1c-2H**, even if there would be a contribution from the open-shell form (Fig. S21). The CASSCF calculations also indicated that the biradical character is negligibly small in **1c-2H** as well as (Ar₄QD)₂S **1** (Fig. S48), which is in good agreement with experimental results."

4. The aromaticity of the dinaphthothiophene (DNT) in different forms could be further elucidated by analyzing the ACID plots and NICS values for the three different forms of **1a**²⁺, **1b**²⁺, and **1c**²⁺.

=> As pointed out by the reviewer, we conducted calculations for anisotropy of the induced current density (ACID) plots (R. Herges, D. Geuenich, *J. Phys. Chem. A* **2001**, 105, 3214. and D. Geuenich, K. Hess, F. Kohler, R. Herges, *Chem. Rev.* **2005**, 105, 3758.) and the nucleus-independent chemical shift (NICS) values.

For ACID plots (isovalue: 0.02) of the three different forms (**A**, **B**, and **C**) in **1a**²⁺, **1b**²⁺, and **1c**²⁺, the similar ring currents for each form were observed regardless of the steric effect of the *ortho* substituents. Here, we use the results of **1a**²⁺ for explanation.

In the *para*-quinoid form **A**, a local paramagnetic ring current was observed in the benzene ring of the central skeleton, and a global paramagnetic ring current was observed over the thienonaphthalene moiety. In the σ -bond form **B**, a global paramagnetic ring current was observed over the heteroacene core. In the *ortho*-diphenoquinoid form **C**, a local paramagnetic ring current was observed on the two benzene rings at both sides.

For NICS values of the three different forms (**A**, **B**, and **C**) in $1a^{2+}$, $1b^{2+}$, and $1c^{2+}$, the similar values for each form were observed regardless of the steric effect of the *ortho* substituents. In order to avoid the influence of the aryl groups, which overlap with the central skeleton, NICS(0) [GIAO B3LYP/6-311+G(2d,p)//CAM-B3LYP-D3/6-31G(d)] was adopted. Here, we use the results of $1a^{2+}$ for explanation.

In the *para*-quinoid form **A**, negative NICS values were observed for the benzene ring and thionaphthalene skeleton, indicating that they have aromatic characters. On the other hand, the six-membered ring of the quinoid moiety showed a slightly positive NICS value, suggesting that the quinoid skeleton is non-aromatic. In the σ -bond form **B**, negative NICS values were observed for whole skeletons of heteroacene core, while strong aromaticity was observed in the two benzene rings at both sides. On the other hand, the six-membered ring containing the σ -bond showed a positive NICS value, suggesting that it is non-aromatic. In the *ortho*-diphenoquinoid form **C**, a negative NICS value was confirmed for the benzene rings, indicating that they are aromatic. In addition, the thiophene ring showed a slightly negative value, implying that it would be aromatic. Considering that the thienoquinoid ring is not an aromatic skeleton, this negative value would be caused by the overlap of congested aryl groups on the diarylmethylene moieties. Indeed, X-ray and theoretical analyses revealed that there are aryl groups on the thienoquinoid skeleton, by which a shielding effect could not be ignored. On the other hand, the six-membered ring of the quinoid moiety showed a

slightly positive NICS value, suggesting that it is non-aromatic. As described above, aromatic and non-aromatic rings were characterized in each form.

These results were added to the revised Supporting Information (Fig. S49 and S50) and the details are as follows:

"Calculations for anisotropy of the induced current density (ACID) plots^{14,15} were performed. For ACID plots (isovalue: 0.02) of the three different forms (**A**, **B**, and **C**) in **1a**²⁺, **1b**²⁺, and **1c**²⁺, the similar ring currents for each form were observed regardless of the steric effect of the *ortho* substituents. Here, we use the results of **1a**²⁺ for explanation. In the *para*-quinoid form **A**, a local paramagnetic ring current was observed in the benzene ring of the central skeleton, and a global paramagnetic ring current was observed over the thienonaphthalene moiety. In the σ -bond form **B**, a global paramagnetic ring current was observed over the heteroacene core. In the *ortho*-diphenoid form **C**, a local paramagnetic ring current was observed on the two benzene rings at both sides."

"To gain insight into aromaticity, nucleus-independent chemical shifts (NICS) calculations^{16,17} were performed for dications. For NICS values of the three different forms (**A**, **B**, and **C**) in **1a**²⁺, **1b**²⁺, and **1c**²⁺, the similar values for each form were observed regardless of the steric effect of the *ortho* substituents. In order to avoid the influence of the aryl groups, which overlap with the central skeleton, NICS(0) was adopted. In the *p*-quinoid form **A**, negative NICS values were observed for the benzene ring and thienonaphthalene skeleton, indicating that they have aromatic characters. On the other hand, the six-membered ring of the quinoid moiety showed a slightly positive NICS value, suggesting that the quinoid skeleton is non-aromatic. In the σ -bond form **B**, negative NICS values were observed for whole skeletons of heteroacene core, while strong aromaticity was observed in the two benzene rings at both sides. On the other hand, the six-membered ring containing the σ -bond showed a positive NICS value, suggesting that it is non-aromatic. In the *o*-diphenoid form **C**, a negative NICS value was confirmed for the benzene rings, indicating that they are aromatic. In addition, the thiophene ring showed a slightly negative value, implying that it would be aromatic. Considering that the thienoquinoid ring is not an aromatic skeleton, this negative value would be caused by the overlap of congested aryl groups on the diarylmethylene moieties. Indeed, X-ray and theoretical analyses revealed that there are aryl groups on the thienoquinoid skeleton, by which a shielding effect could not be ignored. On the other hand, the six-membered ring of the quinoid moiety showed a slightly positive NICS value, suggesting that it is non-aromatic. As described above, aromatic and non-aromatic rings were characterized in each form. "

5. Since quinoidal structures are typically associated with open-shell diradical forms, what are the ground states and diradical characters of **1a**, **1b**, and **1c**?

=> As described above, some quinodimethane (QD)-based systems exhibit open-shell character based on the resonance structures (e.g., *Acc. Chem. Res.* **2017**, *50*, 977–987 [DOI: 10.1021/acs.accounts.7b00004], *J. Am. Chem. Soc.* **2020**, *142*, 5408-5418 [DOI: 10.1021/jacs.0c01003], and *Angew. Chem. Int. Ed.* **2022**, *61*, e 202205729

[10.1002/anie.202205729]). On the other hand, our systems, such as dibenzo-annulated *para*-quinodimethane, anthraquinodimethane (AQD), derivatives with bulky diarylmethylene units and bisquinodimethane (BQD) analogues, generally do not exhibit open-shell character. Due to the large substituents on the exomethylene units, AQD and BQD derivatives adopt highly strained folded conformations with closed-shell characters (our previous work: *Chem. Commun.* **2021**, 57, 7201-7214 [DOI: 10.1039/D1CC02260A], *J. Am. Chem. Soc.* **2021**, 143, 3306-3311 [DOI: 10.1021/jacs.1c00189], *Chem. Eur. J.* **2023**, 29, e202203899 [DOI: 10.1002/chem.202203899], and *Chem. Eur. J.* **2023**, 29, e202301476 [DOI: 10.1002/chem.202301476]).

Indeed, in this paper, both neutral and dicationic species are all closed-shell compounds, which do not exhibit diradical character even at high temperatures. Moreover, we newly performed CASSCF calculations for our systems as well as model compounds in both neutral and dicationic states (Fig S48), as suggested by another reviewer. As a result, all compounds in this paper do not have diradical character ($y_0 = \text{ca. } 0.1$), even for reference compound **1-2H** that has less steric strain.

Thus, no change was made on the manuscript.

Reviewer #3 (Remarks to the Author):

General comments,

In this paper, the authors designed and synthesized a series compound of $(Ar_4QD)_2S$, in which the redox-active QD units are fused by a thiophene moiety. The two QD units are directly connected, while another bridge was formed by a sulfur atom suppresses the rotation of two units to induce severe steric hindrance between the aryl groups. The configuration allows to modulate the steric repulsion and/or electronic interaction between the two electrophores in specific redox states. The authors have unequivocally determined the structures of these redox states using spectroscopic, voltametric, and X-ray diffraction methods, which revealed that the most stable structure of dicationic state can be precisely controlled by modulating the steric bulk of the substituents on the aryl groups in these molecules. However, the reviewer has some concerns, especially missing completed discussions on the electrochemical transformations as claims by the authors, which should be addressed before this paper can be a potential publication on Nature Communications.

=> We really appreciate the reviewer's valuable comments and recommendations. Based on the comments and suggestions, the manuscript was amended, and the details are described below.

Specific comments,

1. The author claimed that the UV-Vis spectra of **1c-2H** stands in contrast to the other neutral species, $(Ar_4QD)_2S$ **1** and hydride adduct **1b-2H**, which do not exhibit such long-wavelength absorbance. It is suggested to further compare the UV-Vis spectra of **1a**²⁺ and **1c**²⁺, and provide the corresponding discussions. It is obvious that **1a** and **1c** have more similar open-shell structures. Additionally, the authors should explain the shift in the UV-Vis absorbance bands, such as the blue shift of **1b**²⁺.

=> First of all, both neutral and dicationic species are all closed-shell compounds, which do not exhibit diradical character even at high temperatures. Actually, as pointed out by the reviewer, some quinodimethane (QD)-based systems exhibit open-shell character based on the resonance structures (e.g., *Acc. Chem. Res.* **2017**, *50*, 977–987 [DOI: 10.1021/acs.accounts.7b00004], *J. Am. Chem. Soc.* **2020**, *142*, 5408-5418 [DOI: 10.1021/jacs.0c01003], and *Angew. Chem. Int. Ed.* **2022**, *61*, e 202205729 [10.1002/anie.202205729]). However, our systems, such as dibenzo-annulated *para*-quinodimethane, anthraquinodimethane (AQD), derivatives with bulky diarylmethylene units and bisquinodimethane (BQD) analogues, generally do not exhibit open-shell character. Due to the large substituents on the exomethylene units, AQD and BQD derivatives adopt highly strained folded conformations with closed-shell characters (our previous work: *Chem. Commun.* **2021**, *57*, 7201-7214 [DOI: 10.1039/D1CC02260A], *J. Am. Chem. Soc.* **2021**, *143*, 3306-3311 [DOI: 10.1021/jacs.1c00189], *Chem. Eur. J.* **2023**, *29*, e202203899 [DOI: 10.1002/chem.202203899], and *Chem. Eur. J.* **2023**, *29*, e202301476 [DOI: 10.1002/chem.202301476]). Moreover, we newly performed CASSCF calculations for our systems as well as model compounds in both neutral and dicationic states, as suggested by

another reviewer. As a result, all compounds in this paper do not have diradical character, even for reference compound **1-2H** that has less steric strain.

Based on the fact that both dications **1a²⁺** and **1c²⁺** are closed-shell species, we describe a comparison of the UV/Vis spectra of these dications. According to the results of ¹H NMR and X-ray measurements, dication **1a²⁺** adopts the *p*-QD form **A**. As described in the manuscript, it was also confirmed because a characteristic charge-transfer (CT) transition from the HOMO on the *p*-QD skeleton to the LUMO on the diarylmethylum units was observed in the NIR region (at around 970 nm). On the other hand, although it was difficult to identify the molecular structure of dication **1c²⁺** by X-ray analysis, DFT calculations suggest that the most stable structure of dication **1c²⁺** is the *p*-QD form **A**. While the redshift of the NIR absorption band was observed for **1c²⁺** in CH₂Cl₂ compared to that for **1a²⁺**, it can be accounted for by the structural difference. According to our previous study, such CT transition can be tuned by the *ortho* substituents on the aryl groups (*Chem. Eur. J.* **2023**, *29*, e202203899 [DOI: 10.1002/chem.202203899]). The larger the *ortho* substituent group, the smaller the dihedral angle between the central skeleton and the diarylmethylum units, and the lower the LUMO level. In this case, the methyl group is larger than the fluorine atom, so that the CT band was redshifted in **1c²⁺**. However, since we were not able to determine the structure of **1c²⁺** by X-ray analysis, we have excluded such speculation in the previous manuscript. Based on the reviewer's comments, the estimation of the redshift was added to the revised Supporting Information.

On the other hand, the blueshift of the HOMO-LUMO transition was observed in **1b²⁺** because **1b²⁺** adopts the σ -bond form **B**. While the coefficients of LUMO were attributed to the electron-deficient diarylmethylum units, which all dications have, those of HOMO were located on the less-donative heteroacene moiety in **1b²⁺**. Thus, the HOMO-LUMO transition was blueshifted in **1b²⁺** with the σ -bond form **B** compared to **1a²⁺** and **1c²⁺** with the *p*-QD form **A**. Therefore, the sentence was amended as follows:

"In comparison, in the case of **1b²⁺**, the NIR absorption band is significantly blue-shifted relative to that of **1a²⁺** and is accompanied by an increase in the corresponding molar extinction coefficient ϵ [λ_{\max}/nm (log ϵ) 698 (4.35)]. This would be caused by the difference in the HOMO level, indicating that **1b²⁺** has less donating π -skeleton such as heteroacene moiety in the σ -bond form **B**. These results suggest that dication **1b²⁺** adopts a structure that is substantially different from the *p*-quinoid form **A** of **1a²⁺**."

2. The redox properties of **1a** and **1c-2H** were characterized by CV curves. However, the CV data of **1b-2H** and its explanation are missing.

=> We conducted the cyclic-voltammetry experiments for **1b-2H** (at 298 K in CH₂Cl₂ containing 0.1 M [Bu₄N⁺][BF₄⁻] as the supporting electrolyte). The voltammogram showed a one-wave two-electron oxidation peak at +1.19 V to give σ -bond-dissociated **1b-2H**²⁺. As a result of the change in structure, the corresponding reduction peak appeared at the far cathodic region (+0.66 V) to produce σ -bonded **1b-2H**. Such a large separation between the oxidation and reduction potentials is a characteristic of dynamic redox (*dyrex*) systems that involve structural changes (reorganization of the electronic structure) associated with redox reactions. The important thing is that both redox waves were observed in almost the same region as the redox potential of dication **1b**²⁺, which has the same skeleton as **1b-2H**. Thus, it was found that both systems have characteristic electronic structures that induce the breaking/formation of the central σ -bond associated with redox reactions. These results were added to the revised Supporting Information (Fig. S17).

3. It is claimed that AQD **I-a** is unlike **1a**, a different structural preference for a folded geometry in the neutral state and a twisted geometry in the dication state. What is the neutral state and twisted geometry of ADQ **I-a**?

=> AQD derivatives generally adopt folded geometry as the most stable structure in the neutral state with and without ortho substituents (our previous work: *Chem. Commun.* **2021**, 57, 7201-7214 [DOI: 10.1039/D1CC02260A] and *Chem. Eur. J.* **2023**, 29, e202203899 [DOI: 10.1002/chem.202203899]). Indeed, X-ray analysis revealed that AQD **I-a** adopts folded geometry (Fig. S25). In addition, when DFT calculations were performed with twisted geometry as an initial structure, twisted geometry was obtained, but it was a transition structure with one imaginary frequency. Thus, neutral species prefer folded geometry.

On the other hand, in the dicationic state, diarylmethyl cation units were generated and orthogonally attached to the central π -skeleton in almost all cases. Thus, dicationic species prefer twisted geometry.

However, we observed reversible redox waves in **1a**, indicating that there is no structural change between neutral/dicationic and dicationic/tetracationic species. According to our

study to date, some derivatives have a contribution of twisted species with a higher HOMO level even though such twisted geometry is a metastable structure (e.g., *Angew. Chem. Int. Ed.* **2024**, *63*, e202316753 [DOI: 10.1002/anie.202316753] and *Chem. Eur. J.* **2024**, *30*, e202400916 [DOI: 10.1002/chem.202400916]). Based on another finding that cationic species only adopt twisted geometry in all cases, we considered that the neutral species with twisted geometry was oxidized in the solution of **1a**. The word, especially for unlike **1a**, could be misleading, so we have revised it as follows;

"A large separation of the redox peaks is observed for the anthraquinodimethane derivative ($E_{\text{peak}}^{\text{ox}} = +1.30 \text{ V}$ and $E_{\text{peak}}^{\text{red}} = +0.71 \text{ V}$; Fig. S41) because it has a different structural preference for a folded geometry in the neutral state and a twisted geometry in the dication state. As the electrochemical measurements suggested that the neutral or dicationic species of **1a** would adopt a geometry similar to its dicationic or neutral state, we aimed to isolate the dication salt of $(\text{Ar}_4\text{QD})_2\text{S } \mathbf{1a}$ and determine its structure."

4. The authors used diverse electrochemical transformations to realize the Para-Quinoid, σ -Bond, and Ortho-Diphenoquinoid forms. However, the electrochemical experiments are merely the CV data for presenting the redox properties of different products. How did electrochemical transformations emerge?

=> Since dications **1a**²⁺, **1b**²⁺, and **1c**²⁺ can be generated by electrochemical oxidation of **1a**, **1b**, and **1c**, we used the word "electrochemical" in the previous version. However, as pointed out by the reviewer, **1c-2H** cannot be obtained by only electrochemical stimulation but by two-electron oxidation of **1c** followed by hydride reduction of the intermediary dication. Thus, the word "electrochemical transformation" was not scientifically appropriate in this paper. Instead of the word "electrochemical transformation," the manuscript, including the title, was amended with the word "redox-mediated transformation."

5. There are significant data in the SI appendix. The sequence of text does not align well with the order of the images of SI appendix. For instance, in the main text, the authors mentioned the data of S41. While it is difficult for the readers to find such a data. It is suggested to carefully arrange the data in main text and SI appendix.

=> Based on the reviewer's comment, Supporting Information was amended appropriately. The order of Experimental Section and Theoretical Study in supporting information was changed to make it easier for readers to understand.

<Addressing the comments>

Reviewer #1 (Remarks to the Author):

The changes made by the authors in the manuscript are almost fully adequate and the clarity of the paper is improved, especially by adding explanation of the effect of steric distortion on the co-planarity of the participating π -subsystems and the subsequent effect on the HOMO energy level and thus, oxidation potential.

=> We are deeply grateful for the reviewer's valuable comments and suggestions. Based on the suggestions, the manuscript was amended, and the details are described below.

As final and crucial suggestions, we state the following:

- Page 9, last paragraph: the statement about the 1c-2H compound having the negligibly small open-shell character is not rigorously correct. In the presented set of compounds, for which diradical characters were calculated from the CASSCF results, all of them are around $y_0 = 0.1$ and higher. This is an important degree of open-shell character for the reactivity. It is more appropriate to state that open-shell character is small, but non-negligible in terms of chemical reactivity, even if this character is not detectable with some of the experimental methods. If authors need to categorize the degree of open-shell character they can pre-define what small and medium diradical characters mean in the context.

Please correct all the instances of such use of the term, because for the readers in this field, such wording will cause confusion.

=> We appreciate the reviewer's careful suggestion. Based on that, the term "negligibly" was removed from the manuscript to avoid misleading.

- Page 11, end of Conclusions: "geometry of the π -conjugation" is an ill-defined phrase. More appropriate to use: "topology of the π -conjugation"/"topology of the π -conjugated system"/"geometry of the π -conjugated system". The preferred term when one refers to the graph relation between overlapping/crossing/parallel π -chains is "topology" rather than geometry. This is because one can achieve equivalent topologies of the π -systems with different geometries, but two different topologies of the π -system cannot be achieved by the same geometry, by definition.

=> Based on the comment, the word "geometry" was replaced with "2D/3D structure or topology".

Reviewer #2 (Remarks to the Author):

The authors have addressed all of my questions. I recommend the acceptance of this manuscript for publication.

=> We sincerely appreciate the reviewer's recommendation.

Reviewer #3 (Remarks to the Author):

The authors have addressed the raised concerns. This work can be a potential publication on Nature Communications.

=> We sincerely appreciate the reviewer's recommendation.

Recommendation: This manuscript is adequate for the publication, but should be revised by authors based on reviewer feedback and modified version should be reviewed again before the manuscript is finally accepted.

Important remarks and suggestions:

The article by Harimoto et al. exemplifies how co-dependent electronic and molecular structure are tuned by the presence of steric hindrance.

By increasing the steric hindrance from hydrogen, to fluorine, to methyl attached to the aryl groups of diarylmethylene terminal units of sulfur bridged quinodimethane derivative, the multiconfigurational electronic structure is tuned.

The ground state electronic wavefunction of presented compounds (**1a**, **1b**, **1c**) is the combination of multiple valence bond forms (VBFs, i.e. resonance structures) of the π -conjugated subsystem.

The relative contribution of each VBF depends on the specific build of the molecule as changing some groups would induce the compound to assume significantly different geometry. The relative contribution of each VBF in the compounds is implied by authors based on the analysis of the geometry given by x-ray crystallography and is further corroborated by the electrochemical behavior of **1a**, **1b**, **1c** and structures/properties of their dication species.

The article is well presented, accessible to chemists from broad backgrounds and is interesting for the researchers in various subfields of molecular chemical science. Nevertheless, the article needs a revision to make some improvements in insights, fill the gaps in discussion and correct a few technical points.

- The quality of the three-dimensional drawings of the crystal structure should be improved as sometimes the relevant structural details that would lead to the specific features of the electronic structure are elusive. Furthermore, the authors need to discuss the effect of dihedral angles between fragments of the molecule more rigorously. For example, the π -conjugated diarylmethylene fragments are not fully coplanar with the π -conjugated core of molecules (Tables S3-S5) and thus, the overlap between p orbitals at the interface of different fragments is not optimal for π -bonding. This obviously affects the electronic structure. Therefore, this analysis should be integrated into the main text concisely but more explicitly.
- Electronic structure problem should be more precisely defined for the explored species. Hence, rigorous theoretical analysis based on resonance theory/valence bond theory within a multiconfigurational quantum chemistry formalism serves useful. Also, the description of the electronic structure by DFT methods is rather simplistic and insufficient for the appropriate theoretical and computational analysis.

Firstly, it is crucial to study the simplified model of the presented compound set, which would reveal the electronic structure based on the topology of π -conjugation.

One can use the methylene linker to avoid steric hindrance and maintain the same topology of π -conjugation.

This is why the model system below is important to analyze.

Note that π -bonds can be rearranged from the drawn, but the overall topology of π -conjugation is defined by tailoring the geometry.

The electronic structure of this model system should be analyzed as a reference for the compounds **1a**, **1b**, **1c**, to better understand how these different groups modulate the properties initially determined by the topology of π -conjugation in the case of negligible steric hindrance. This will improve the insight into the rational design principles.

Secondly, DFT description of such multiconfigurational compound is usually incomplete since, in addition to being single-reference method, DFT shows significant dependence of energy gaps and properties on the choice of exchange-correlation functional.

In order to better understand the electronic structure of this multiconfigurational systems, it is highly recommended to use multireference methods such as complete active space self-consistent field (CASSCF) or at least complete active space configuration interaction (CASCI) with proper guess orbital set.

The active space for this electronic structure problem is not too large and is computationally tractable with common computing clusters (probably CAS(14,14) is the sufficient size of active space and moderate-sized basis set is enough), as aryl groups in diarylmethylene parts do not participate in the subspace of frontier π -orbitals. The authors can use DFT-optimized geometries and for each geometry (different DFT-optimized forms) for each compound, determine the excited states by CASSCF/CASCI method.

The results can be analyzed in terms of CASSCF natural orbitals and their occupation numbers to obtain the description of electron density distribution of most relevant part of orbital set for a given electronic problem.

In addition, the contents of the Table S1 and Table S2, describing the different reference system (**1a'**-**1c'** and **1a''**-**1c''**, and **1c-2H''**) should be more carefully discussed and may have to be re-formulated and re-evaluated. The important difference between **1a''**, **1b''**, **1c''**, **1c-2H''** (second set of chosen reference systems) and **1a**, **1b**, **1c**, (primary systems) is that compounds **1a''** to **1c''** and **1c-2H''** do not have terminal fused benzene rings as opposed to **1a**, **1b**, **1c**, that significantly affects the electronic structure.

That is why the diradical character would much be higher in the case of **1c-2H''** compared to **1c-2H** because shifting from closed-shell to open-shell form allows to gain

two Clar's π -sextets for **1c-2H''**, while there is no such change for **1c-2H** (also see comment 2 below).

Note that increased steric hindrance between aryl groups in the presented systems would distort the internal six-membered ring (Tables S3-S10) so that the aromatic resonance energy would be noticeably diminished compared to benzene and terminal benzenoid rings.

The statement that **1c-2H** is singlet diradical (row 2 from below and column 5 of Table S2) seems to contradict the statement in the main text, page 9 (also see comment 2). Authors must correct **1c-2H** by **1c-2H''** (second last row of Table S2) as this error causes confusion in the reader.

Also, it is not fully clear what authors mean by singlet diradical and which criteria are applied for such categorization.

As mentioned, the analysis of CASSCF and/or CASCI results would give us more detailed and qualitatively accurate description. This is necessary to determine the degree to which the electronic structure of the given state is singlet diradical.

Therefore, analysis of diradical character based on the frontier natural orbital occupation numbers and description of excited states by CASCI/CASSCF in such cases is crucial.

Comments:

1. Figure 1. The electronic structure of forms A, B, C may not be clear to the general reader: Is it a diradical, dication or closed-shell?

It is highly desirable that this is visually better clarified in the chemical drawings of compounds.

2. "Moreover, even at higher temperatures in DMSO-*d*6, sharp NMR signals were observed, which suggests that the open-shell character in **1c-2H** is negligible, even though in the open-shell form, resonance energy can be gained by aromatization of the two rings (Fig. S40). " – page 9.

If one analyzes the valence bond forms properly, one sees that this reasoning is based on inaccurate assumption.

In the simplified representations of the compound **1c-2H** (this applies to **1a**, **1b** and **1c**, analogically too), the count of Clar's sextets remains the same irrespective of closed-shell or open-shell structure.

This is exactly the reason why the open-shell character is negligible, because there is no significant resonance energy gain in open-shell forms (two Clar's sextets) compared to closed-shell form (two Clar's sextets).

This must be addressed in the revised version of the article by more rigorous analysis suggested here.

Moreover, as mentioned above, the six-membered rings bridged by sulfur are geometrically more distorted than the terminal six-membered rings.

Hence, the aromatic resonance energy of the terminal six-membered rings is expected to be larger compared to internal six-membered rings upon assuming the Clar's π -sextet configuration.

Since the closed-shell structure has Clar's π -sextets in the terminal rings, there is no significant gain (if any) in aromatic resonance energy upon opening the shell and thus, the open-shell character remains negligible.

Therefore, the correction must be made in the main text and as mentioned above, more careful comparison between presented primary compounds and assumed reference compounds is necessary with the new reference compound that we suggested above.

3. "Thus, this study is the very first example of the successful modulation/control of different pathways toward electrochemically and/or thermodynamically stable structures in multi-electrophore systems. This work can be expected to lead to the development of a library of unprecedented π -electron compounds." – page 11.

Even though the modulation of assumed ground-state geometry as a function of steric hindrance is remarkable and is quite well explained, in order to generate the large library of important π -electron compounds, one needs to study the influence of other factors such as topology of π -conjugation and electronic effects (electron-donating/withdrawing).

The modulation of structures only as a function of steric effects does not necessarily allow the desired degree of freedom in rational design.

In π -conjugated compounds, the principal factor that controls the properties is the topology of the π -conjugation.

One can state that steric effects are final fine-tuning of the co-dependent geometry and electronic structure of the π -conjugated compound, as the topology of the π -system sets the constraints and defines the nature of contributing valence bond forms in the electronic structure, contribution of which can then be modulated with adjusting the steric hindrance.

Thus, it would be more appropriate to moderate the final sentence of the article with more specific statement, such as stating the mechanism of control of structures in multi-electrophore systems and the specific degree of freedom it allows in the presented π -conjugated systems, rather than ambiguously stating that this approach can be expected to lead to an unprecedented library of π -electron compounds.

In other words, authors should specify the degree of variety you could expect in the library of π -electron compounds because based on the steric effects alone, one cannot generate a significant portion of chemical space of π -conjugated compounds with a sufficient variety of properties.

Even though this does not invalidate the presented work, the concluding statement is exaggerated and should not go outside the scope of the article.

The completed experimental work seems well-performed for the study of properties and provides very useful data for the analysis of controlling the preferred form of the sulfur-bridged quinodimethane derivatives.

To summarize, as a theoretical reviewer, I would still strongly recommend that one of the reviewers of this manuscript has expertise in the used experimental methods and experience in structural determination or synthesis of the presented or related class of compounds.